

# Developing a hydrological monitoring and sub-seasonal to seasonal forecasting system for South and Southeast Asian river basins

Yifan Zhou[1], Benjamin F. Zaitchik[1], Sujay V. Kumar[2], Kristi R. Arsenault[2,3], Mir A. Matin[4], Faisal M. Qamer[4], Ryan A. Zamora[1], Kiran Shakya[4]

[1]Department of Earth and Planetary Sciences, Johns Hopkins University, Baltimore, Maryland, USA.
[2]Hydrological Sciences Laboratory, NASA Goddard Space Flight Center, Greenbelt, Maryland, USA
[3]Science Applications International Corporation, Reston, VA, USA
[4]International Centre for Integrated Mountain Development, Kathmandu, Nepal

*Corresponding to*: Benjamin F. Zaitchik (zaitchik@jhu.edu)

**Abstract.** South and Southeast Asia is subject to significant hydrometeorological extremes, including drought. Under rising temperatures, growing populations, and an apparent weakening of the South Asian monsoon in recent decades, concerns regarding drought and its potential impacts on water and food security are on the rise. Reliable sub-seasonal to seasonal (S2S) hydrological forecasts could, in principle, help governments and international organizations to better assess risk and act in the face of an oncoming drought. Here, we leverage recent improvements in S2S meteorological forecasts and the growing power of Earth Observations to provide more accurate monitoring of hydrological states for forecast initialization. Information from both sources is merged in a South and Southeast Asia sub-seasonal to seasonal hydrological forecasting system (SAHFS-S2S), developed collaboratively with the NASA SERVIR program and end-users across the region. This system applies the Noah-MultiParameterization (NoahMP) Land Surface Model (LSM) in the NASA Land Information System (LIS), driven by downscaled meteorological fields from the Global Data Assimilation System (GDAS) and Climate Hazards InfraRed Precipitation products (CHIRP and CHIRPS) to optimize initial conditions. The NASA Goddard Earth Observing System Model - sub-seasonal to seasonal (GEOS-S2S) forecasts, downscaled using the National Center for Atmospheric Research (NCAR) General Analog Regression Downscaling (GARD) tool and quantile mapping, are then applied to drive 5-km resolution hydrological forecasts to a 9-month forecast time horizon. Results show that the skillful predictions of root zone soil moisture can be made one to two months in advance for forecasts initialized in rainy seasons and up to 8 months when initialized in dry seasons. The memory of accurate initial conditions can positively contribute to forecast skills throughout the entire 9-month prediction period in areas with limited precipitation. This SAHFS-S2S has been operationalized at the International Centre for Integrated Mountain Development (ICIMOD) to support drought monitoring and warning needs in the region.



## 1 Introduction

South and Southeast Asia is one of the most populated areas in the world, and a significant portion of livelihoods depend directly or indirectly on smallholder agriculture. Agriculture is one of the most weather-dependent human activity (Hatfield et al., 2011), and smallholder systems are particularly vulnerable to weather variability, including extreme events such as drought. South and Southeast Asia have been experiencing anthropogenic warming since the 1950s (Sivakumar and Stefanski, 2010), and the warming is projected to continue in the near future (Barros and Field, 2014). The frequency of extreme weather events,

including droughts, has also been increasing under this warming trend, with implications for food security and social stability in a conflict-prone region that already includes extensive marginal agriculture on semi-arid lands (Samaniego et al., 2018). Subseasonal-to-seasonal hydrological forecast systems (S2S-HFS) have the potential to aid preparedness for these extreme events. Such systems have been implemented all over the world at the scale of large river basins (Yuan et al., 2016;Getirana et al., 2009), countries (Xia et al., 2012;Bell et al., 2017;Shah et al., 2017), continents (Wanders et al., 2019;Sheffield et al.,

2014;Yuan et al., 2013) and the entire globe (Alfieri et al., 2013;Yuan et al., 2011;Wanders and Van Lanen, 2015). The forecast period has been increased from sub-monthly to monthly scale (Thielen et al., 2009;Alfieri et al., 2013) in early years to six to nine months (Wanders et al., 2019;Arsenault et al., 2020) in recent years.

An S2S-HFS generally includes three components—a meteorological forecast, a downscaling method, and a hydrological / land surface model (Yuan et al., 2015;Hao et al., 2017). The presence of a land component in this system means that

hydrological forecasts can draw skill from both the quality of meteorological forecast and the accuracy of modelled initial hydrological states. The influence of initial hydrological states has a more substantial impact during the early prediction period, while meteorological forecast dominates later months (Shukla and Lettenmaier, 2011). The influence of initial hydrological states has a more substantial impact during the early prediction period, while meteorological forecast dominates later months (Shukla and Lettenmaier, 2011). The initial condition, however, can still positively contribute to the forecast skill several

months after the initialization (Samaniego et al., 2019). This contribution can come from the memory of deeper soil moisture, groundwater storage and cumulated snow pack in earlier seasons, which provides the potential for hydrological forecasts to have skills greater than meteorological S2S forecasts (Koster et al., 2010).

South and Southeast Asia present both a challenge and an opportunity in this regard. The challenge is that S2S meteorological forecasts can be quite difficult in some areas. For example, de Andrade et al. (2019) study the skill of meteorological S2S

forecasts and find that for multiple initialization dates, the forecasts lose meaningful skill within four weeks of initialization. Similarly, Jie et al. (2017) conclude that the operational climate models from the WCRP Seasonal to Subseasonal Prediction Project lack the skill for prediction of the Asian summer monsoon, a major precipitation source of most areas in South Asia, beyond one month in advance. The opportunity is that many rivers in major basins of South and Southeast Asia are sourced from mountainous regions with significant snowpack and seasonally frozen soil water, such that a properly initialized

hydrological forecast has the potential to add considerable predictive power beyond the meteorological forecast.



In this study, we aim to establish an operational fine-resolution, continental hydrological monitoring and forecasting system for South and Southeast Asia: The South and Southeast Asia Hydrological Forecast System-S2S (SAHFS-S2S). This system includes downscaled meteorological fields from a dynamically-based S2S meteorological forecast system, an advanced land surface model, and post-processing tools to derive drought indicators. The system aims to provide end-users in South and
Southeast Asia with water resources information to help manage local drought risks and strengthen food security.

We first test the forecasting system in hindcast experiments against a retrospective run of the monitoring system, which is our best estimate of hydrological states, and available satellite-derived estimates of soil moisture, the critical variable for our drought monitoring goals. An additional control set of hindcast simulations are designed to evaluate the importance of initial conditions to the hydrological forecast skill. All settings in this set of control hindcast simulations are the same as the forecast
system except that the initial conditions are set as the same climatological hydrologic states averaged from the entire hindcast period.

Section 2.1 describes the precipitation pattern in research areas and major river basins. Section 2.2 describes the dataset used in this study. Details of the monitoring and forecasting system are described in section 2.3 and section 2.4, respectively. The post-processing of both systems is described in section 2.5. The comparisons among different simulations, the importance of
the initial conditions, and validation of both systems and applications of the systems are discussed in section 3. Finally, the implications, limitations, and possible future work are discussed in section 4.

## 2 Methods

### 2.1 Research area and precipitation patterns

In this study, all simulations are performed on a domain within the South and Southeast Asia region, ranging from 8 °N to 45
°N and 58 °E to 123 °E. The analysis, however, focuses on a study domain, mainly including five major river basins and surrounding areas (Fig. 1a). The five major river basins in this South and Southeast Asia study domain are the Helmand, Indus, Ganges, Brahmaputra, and Mekong basins.

Among these five river basins, the Mekong, Brahmaputra, and Ganges have the highest average precipitation, annually: 1677mm, 1227mm, and 1108mm, respectively. These three basins are strongly influenced by the South Asian monsoon pattern,
and most precipitation falls in summertime monsoon season (80% in the Mekong basin, 70% in the Brahmaputra basin, and 85% in the Ganges basin). The monsoon seasons generally start in May, June, and July in the Mekong basin, the Brahmaputra basin, and the Ganges basin, respectively, and ends in late September or early October. The Mekong and Brahmaputra basins have considerable precipitation variations between the upper and lower basins. Moisture that enters the basins from the south is naturally blocked by the Himalaya ranges and cannot reach the upper basins of these river systems. The upper Brahmaputra
basin suffers from a direct rain shadow effect of the Himalayas and has precipitation amount less than one-fourth of the precipitation in the lower Brahmaputra basin (Immerzeel, 2008). Although the monsoon precipitation falls mostly in the lower





Mekong basin, snow accumulated in the upper Mekong basin provides vital water resources for the entire basin during pre-monsoon and dry seasons (Frenken, 2012).

In the Indus basin, the seasonal cycle is also influenced by the monsoon climate (Chen et al., 2016). Most precipitation falls

during the monsoon season from July to September (199mm), accounting for 51% of the annual average precipitation (389mm). The pre-monsoon and winter seasons, however, also contribute an essential part of the precipitation, while the post-monsoon period is the driest season. Despite providing less total precipitation than the summer monsoon, winter precipitation is vital for wheat and barley crops. Indus also has significant spatial precipitation variation. Most precipitation falls in the mountainous regions in the upper basin, while the lower basin lies in one of the driest areas in South Asia. Melting water from

snow and glaciers in these mountain ranges provides a considerable amount of fresh water to the Indus river (Fowler and Archer, 2006).

The Helmand basin has hyperarid to arid climate (Whitney, 2006), with an annual average precipitation of only 144mm. The seasonal cycle of precipitation is different from the other four basins, as the maximum precipitation occurs during the winter season, which accounts for 76% of the total precipitation. The Helmand basin also suffers from high temperature and extreme

wind which intensify the aridity in the basin.

## 2.2 Data

Datasets used in this research include the daily Climate Hazards Center InfraRed Precipitation data (CHIRP) and Climate Hazards Center InfraRed Precipitation with Station data (CHIRPS), National Oceanic and Atmospheric Administration (NOAA)'s Global Data Assimilation System (GDAS), NASA's Goddard Earth Observing System Model sub-seasonal to

seasonal forecast version 1 and version 2 (GEOS-S2S-V1 and V2), and the European Space Agency's Climate Change Initiative for Soil Moisture (ESA-CCI SM).

CHIRP is a precipitation dataset developed by Climate Hazards Center at University of California, Santa Barbara. This dataset has a quasi-global (50 °S - 50 °N and all longitude) spatial coverage and is derived from satellite data (Funk et al., 2015). The data is available from 1981 to near real-time present with a 2-3-day latency. This data has 0.05-degree spatial resolution, and

the daily precipitation product is used in this research. CHIRPS is a precipitation dataset derived by combining the CHIRP dataset with in-situ station data (Funk et al., 2015). The specifications of the dataset are similar to CHIRP, except that this dataset has about 3-week latency. In this study, the CHIRP/CHIRPS dataset provides the precipitation field in the monitoring system (see section 2.3 for details) and CHIRPS is used as the reference for downscaling precipitation forecasts in the forecasting system (see section 2.4 for details).

GDAS is an atmospheric analysis system developed at the National Center for Environmental Prediction (NCEP) at NOAA (National Climatic Data Center, 2020). It is produced by assimilating surface observations into a Global Forecast System (GFS). The assimilated surface observation includes balloon data, wind profiler data, aircraft reports, buoy observations, radar observations, and satellite observations (https://www.ncdc.noaa.gov/data-access/model-data/model-datasets/global-data-assimilation-system-gdas). The GDAS dataset has a 6-hourly temporal resolution and produces short-term meteorological



forecasts at 0-hour, 3-hour, 6-hour and 9-hour lead time. GDAS data are available from 2000 forward and have real-time
updates. The spatial resolution starts at about 1° × 1° in the year 2000 and has gradually improved to an equivalent grid of
0.125° × 0.125°, since early 2015. In this study, seven meteorological variables from GDAS are used as forcings in the
monitoring system (see section 2.3 for details) and baselines for meteorological downscaling in the forecasting system (see
section 2.4 for details). These variables include downward long-wave radiation, downward shortwave radiation, air
temperature, specific humidity, air pressure, zonal and meridional wind speed fields.

GEOS-S2S-V1 is a meteorological forecast dataset produced by the GEOS atmospheric model at the Global Modeling and
Assimilation Office (GMAO) at NASA Goddard Space Flight Center. GEOS-S2S-V1 forecasts were initialized about every
five days in hindcast experiments and forecasts from 2000 to 2017 and produce daily forecasts for nine months. In each month,
one forecast initialization date contains ten ensemble members while others only have one. GEOS-S2S-V1's spatial resolution
is 1°×1.25°.

Eight variables from daily GEOS-S2S-V1, including the same seven variables as in GDAS and precipitation rate variables (the
eighth variable), are used in this research.

ESA-CCI SM is a global gridded surface soil moisture dataset derived from remote sensing observation (Gruber et al., 2019).
The ESA-CCI SM contains three sub-datasets: ACTIVE dataset, PASSIVE dataset, and COMBINED dataset. The ACTIVE
dataset is derived by merging satellite datasets measured by active scatterometer instruments. The ACTIVE dataset spans from
August 1991 to December 2019. The PASSIVE dataset is derived by merging satellite datasets measured by passive radiometer
instruments. The PASSIVE dataset spans from November 1978 to December 2019. The COMBINED dataset is merged and
rescaled from the ACTIVE and the PASSIVE dataset. All three sub-datasets have a daily temporal resolution and 0.25° spatial
resolution. The COMBINED dataset is used in this study to evaluate the hydrological monitoring and forecasting system.

### 2.3 Monitoring system

The monitoring system is an instance of a Land Data Assimilation System (LDAS), which is a technique that merges
observations with physically-based models to produce optimal estimates of terrestrial hydrological states and fluxes (Rodell et
al., 2004;Mitchell et al., 2004). For purposes of monitoring hydrological extremes, our process consists of three steps (Fig. 2):
1) meteorological data processing; 2) land surface model simulations; and 3) post-processing to obtain relevant metrics. The
first two steps are completed within NASA's Land Information System Framework (LIS; Kumar et al. (2006)).

Our meteorological estimates consist of the seven GDAS variables listed above, plus precipitation. Total precipitation is
extracted from CHIRPS at a daily scale for retrospective analysis. Daily precipitation is disaggregated to 6-hourly estimates
based on the diurnal cycle of MERRA-2 precipitation, using LIS's Land surface Data Toolkit (LDT; Arsenault et al. (2018)).
The remaining meteorological forcing variables are extracted from GDAS and downscaled to the spatial scale of the monitoring
system (5 km), using bilinear interpolation with lapse-rate and aspect-slope correction within LIS.

The land surface model uses information from meteorological forcing variables to estimate hydrologic variables through a
physically-based representation of hydrologic processes and land surface energy balance. For this study, we present a



monitoring system that uses the Noah-MultiParameterization (Noah-MP) land surface model. Noah-MP is an augmentation of the Noah LSM, which was first implemented in the NCEP Eta Data Assimilation System (EDAS) mesoscale forecast suite and

NCEP Global Forecast System (GFS) to provide land surface feedback to climate models (Ek et al., 2003). The Noah LSM provides feedback by simulating the water fluxes and energy balance among canopy, vegetation, soil, streamflow, and snow (Livneh et al., 2010;Ek et al., 2003). Noah-MP augments the model representation of physical processes in the surface energy balance, snow and frozen soil, groundwater, runoff, and leaf dynamics. In addition, Noah-MP enables multiple options for a variety of physical processes within the model (Niu et al., 2011;Yang et al., 2011).

A variety of spin-up techniques has been used to acquire initial conditions for hydrologic models (Seck et al., 2015) and land surface models (Cai et al., 2014;Nie et al., 2018). Here, a 57-year offline spin-up is performed by simulating the period 2000-2018, three times, to obtain equilibrium hydrological states (e.g., groundwater storage) under prevailing climate patterns for our meteorological data sources. These equilibrium states, in principle, help monitoring simulations reach the best estimation of hydrological outputs once the monitoring period begins. The monitoring system is then initialized on January 1st, 2000,

owing to the availability of the GDAS data product, and is run up to near real-time. The system operates at a 15-minute time step and generates water and energy fluxes and states at a 5-km spatial resolution that we save at daily temporal resolution, including soil moisture, evapotranspiration, terrestrial water storage, snow water equivalent, among others. The system has satellite data assimilation capabilities (Getirana et al., 2020c;Kumar et al., 2020;Xue et al., 2019;Kumar et al., 2019), but the simulations presented here are open-loop simulations in which satellite data are integrated via parameter fields and

meteorological forcings rather than through active land data assimilation.

Satellite-informed input parameters include the 1 km2 resolution Moderate Resolution Imaging Spectroradiometer-International Geosphere Biosphere Program (MODIS-IGBP) land cover dataset (Friedl et al., 2010), 5-minute FAO soil texture dataset (http://iridl.ldeo.columbia.edu/SOURCES/.NASA/.ISLSCP/.GDSLAM/.Hydrology-Soils/.soils/.dataset_documentation.html), 30-meter Shuttle Radar Topography Mission (SRTM) elevation dataset (Farr et al.,

2007), 0.144-degree global albedo maps (Csiszar and Gutman, 1999) and green vegetation fraction maps (Gutman and Ignatov, 1998) derived from measurements made by the advanced very high resolution radiometer (AVHRR) onboard NOAA's polar orbiting satellites. These parameters are represented as climatologies in our simulations.

We ran our monitoring system in retrospective mode from 2000 to 2017 to provide a baseline that is used when evaluating the forecast system. From that time forward, the monitoring system has run in near real-time mode. On account of the 3 week

latency of the CHIRPS product, we use CHIRP precipitation to extend retrospective simulation to real-time. The simulations during this period is re-run with CHIRPS once CHIRPS becomes available.

## 2.4 Forecasting system

The forecasting system applies downscaled ensemble GEOS-S2S forecasts to drive Noah-MP simulations out to a nine-month forecast lead. The workflow and model specifications of the forecasting system are similar to the monitoring system except

for meteorological forcing data processing (Fig. 2).





In the forecasting system, the same meteorological forcing variables as in the monitoring system are extracted from GEOS-S2S data products (see details of GEOS-S2S in section 2b). Due to the coarse resolution and inevitable bias of global climate model outputs, these forcing variables from GEOS-S2S-V1 are downscaled and bias-corrected to corresponding monitoring system forcing variables (i.e., precipitation from CHIRPS and other variables from GDAS) using a Generalized Analog and

Regression Downscaling (GARD) algorithm (https://github.com/NCAR/GARD; Gutmann et al. (2020)). This downscaling algorithm takes a training dataset, prediction dataset, and observation dataset as inputs. The observation dataset contains records of variables (dependent variables) with targeted fine spatial resolution, for example, the precipitation from the CHIRPS dataset. The training dataset includes records of coarse spatial resolution variables (independent variables), which have the same time resolution as the dependent variables, for example, hindcasted GEOS-S2S-V1 precipitation. The prediction dataset

contains the records of the same independent variables as the training dataset but acquired in the forecast period—for example, new GEOS-S2S-V1 forecast precipitation.

In this application, we apply GARD to predict each forecast variable as a function of the same variable in the training datasets (e.g., precipitation to predict precipitation). GARD has the capability to use multiple predictor variables to improve downscaling accuracy, but the influence of the different combinations of independent variables in GARD is beyond the scope

of this paper. The analog-regression downscaling approach in GARD comprises two steps. First, in the analog step, the algorithm searches for a user-defined number of records in the training dataset, the values of which are the most similar to the value in the prediction record to be downscaled. Second, in the regression step, the training records selected in the first step are used to train a linear regression model with observation records in corresponding time steps. The prediction record is fed to this trained linear regression model to calculate the dependent variable at the targeted fine spatial resolution.

GEOS-S2S-V1 products consist of a series of meteorological forecasts initialized about every five days. To construct proper training datasets, we only use the first several days (approximately five days) of each forecast simulation before the next forecast is initialized to create one record of independent variables per day from the year 2000 to 2017. This approach to creating training datasets aims to take advantage of frequent (approximately every five days) GEOS-S2S-V1 hindcasts and to avoid outliers created by mismatched GEOS-S2S-V1 variables with corresponding GDAS/CHIRPS variables due to sharp

forecast skill decline of climate models in later forecast periods (Jie et al., 2017). CHIRPS and GDAS are used to construct the observation dataset. Due to the changing resolution of GDAS data product, coarser-resolution data in the early years are downscaled to 0.125° × 0.125° using bilinear interpolation with lapse-rate and aspect-slope correction to unify the spatial resolution. This 0.125° × 0.125° resolution GDAS product is then aggregated into daily data to unite time intervals with the training dataset.

In order to correct the evolving bias between longer lead-time GEOS-S2S-V1 precipitation hindcasts and CHIRPS, the total precipitation variable is further post-processed using a cumulative distribution function (CDF) matching method (Yuan et al., 2014). In each month within the 9-month forecast period, daily GARD-downscaled GEOS-S2S-V1 precipitation and CHIRPS precipitation from 2000 to 2017 are used to construct the precipitation forecast and observation CDF function separately. We





acknowledge that it is possible to achieve bias correction on a finer time scale by constructing GARD training dataset and CDF

functions at sub-monthly scale (e.g., weekly scale), but shortening the time scale will also reduce the size of the training data. The downscaled meteorological variables are further disaggregated from the daily time scale to the 6-hourly time series to capture sub-daily time variation. Total precipitation and solar radiation are disaggregated using the ratio between 6-hourly and daily GDAS climatology data, while the other six variables are disaggregated using the difference between 6-hourly and daily climatology data.

A set of hindcast simulations has been initialized on May 1st from the year 2000 to 2017. The meteorological forcing of these simulations is GEOS-S2S-V1 initialized on May 1$^{st}$, downscaled following the forecasting system workflow and the initial conditions of these simulations are obtained from retrospective simulations (i.e. "real" initial condition; Fig. 2). Each initialized hindcast simulation has ten ensemble members and lasts for nine months (i.e., May to January of next year). May 1st was chosen as the initialization date of this experiment to capture the monsoon seasons in the monsoon regions and dry season in

the western part of the domain (see Section 2.1 for details). This set of hindcast simulations with "real" initial conditions (hereafter referred to as hindcast-RIC simulations) is designed to evaluate the performance of the forecast workflow.

An additional set of control hindcast simulations with climatological initial conditions (hereafter referred to as hindcast-CIC simulations; Fig. 2) is designed to study the impact of initial conditions on the forecast skill. In this set of hindcast-CIC simulations, the workflow and all settings are the same as hindcast simulations except for initial conditions. All 18 hindcast-

CIC simulations use the climatological initial conditions (CIC) calculated by averaging May 1st hydrological states from 2000 to 2017 obtained from retrospective simulations.

In additional to the hindcast simulations, the forecast system is also running operationally, monthly, with the same forecast workflow but driven by downscaled GEOS-S2S-V2 product instead of downscaled GEOS-S2S-V1 product. GEOS-S2S-V2 dataset also provides 9-month daily meteorological forecasts about every five days, but with finer spatial resolution of

0.5° × 0.5°. Evaluation of the operational forecast system is planned for the future, however, because GEOS-S2S-V2 offers only a small ensemble prior to 2017, making evaluation less reliable at this time, this paper focuses on the evaluation of the hindcast simulations driven by downscaled GEOS-S2S-V1 from 2000 to 2017.

In the hydrological forecasting system, meteorological forecast forcing data are sourced from all ensemble members launched on a common GEOS-S2S forecast initialization date; i.e., the one date per month that offers the ten-member ensemble. It is

possible to include other GEOS-S2S meteorological forecasts in the hydrological forecasting system, but proper bias-correction methods are needed to correct inconsistent forecast skill/bias among GEOS-S2S meteorological forecasts with different initialization dates. For example, in the hindcast-RIC hydrological simulations initialized at May 1st, a combination of April 26th and May 1st GEOS-S2S-V1 meteorological forecasts can be used to enlarge the size of the ensemble members. However, sub-monthly/monthly GARD/CDF post-processing methods would need to be applied to April 26th and May 1st

forecasts separately.


## 2.5 Post-processing

In this study, the evaluation of the monitoring and forecast systems is performed on monthly scales. All simulations are first averaged from daily to monthly timescale. The ensemble mean of forecast simulations generally shows much less variance than retrospective simulations due to averaging effects (Koster et al., 2004). To compare simulations from both systems and
other datasets, all simulation results and data products are standardized before comparison.

Climatological monthly mean and standard deviations for retrospective simulations are calculated for each month separately (January to December) with simulation results from 2000-2017. The retrospective simulation results are then standardized in the form of the standardized anomaly with the following equation,

$$stdamly_{yyyymm} = \frac{val_{yyyymm} - \overline{val_{mm}}}{std_{mm}},$$

where $stdamly_{yyyymm}$ is the standardized anomaly for year $yyyy$ and month $mm$, $val_{yyyymm}$ are the original values from simulations and analysis datasets for year $yyyy$ and month $mm$, $\overline{val_{mm}}$ is the climatology mean for month $mm$ and $std_{mm}$ is the climatological standard deviation for month $mm$. We note that GDAS is an operational analysis system chosen for its low latency. It is not a time-consistent reanalysis product. This means that over the study period GDAS underwent several significant changes in input and data structure, including changes in spatial resolution. All GDAS data were regridded to a
common resolution, and lapse rate and slope-aspect corrections were applied to downscale to topography. Nevertheless, the use of this operational product does mean that anomalies calculated against the long-term mean can contain some statistical artefacts. As the primary purpose of these anomalies is application to drought monitoring, we accept this limitation with the understanding that the system can be used to capture significant drought events but that it is not optimized for trend detection or for precise ranking of event intensities over time.
The forecast simulation results are standardized similarly, but ensemble mean values and separate ensemble member values are standardized separately with respective climatological mean and standard deviation. The climatological monthly standard deviations of separate ensemble values are calculated with all ten ensemble members.

## 3 Results and discussion

### 3.1 Retrospective simulations vs. hindcast-RIC simulations

For purposes of agricultural drought prediction, we are most concerned with our ability to predict soil moisture anomalies. As large-scale networks of root zone soil moisture (RZSM) observations are rare in South and Southeast Asia, we evaluate the prediction skill of the forecast system of SAHFS-S2S by comparing RZSM estimates in hindcast-RIC simulations to RZSM in the retrospective simulations. Since the retrospective run and the hindcast-RIC simulations use the same land surface model, these comparisons aim to evaluate the impact of meteorological forcing on the prediction skill of RZSM. The comparison is
performed by calculating the inter-annual correlation coefficients (R) of monthly RZSM at different lead times, from May (1-



month lead time) to January (9-month lead time). Fig. 3 shows maps of these correlation coefficients across forecast lead times. Higher correlations indicate more skillful RZSM hindcasts. In May (1-month lead time), the correlation between hindcast-RIC and the retrospective simulations is positive in most of the region, and the correlation is significant at 0.95 significant level except for northeast India. As expected, correlations drop in strength and significance as lead time increases. In June (2-month

lead time), the correlation remains positive and significant in the Indochinese Peninsula and the west of the research domain. The prediction skill of the RZSM drops quickly and becomes insignificant in most of the regions except for the westernmost area of the domain two or three months after the forecast initialization date.

We repeat this analysis at the basin scale for the five major river basins of the South and Southeast Asia area (highlighted in Fig. 1). Fig. 4 shows RZSM and precipitation comparison in the Ganges basin, which receives considerable precipitation

during the summer monsoon. Please note that the time axis in Fig. 4 is rearranged, so that data of a specific lead month are grouped. Fig. 4a shows the monthly precipitation time series averaged for the Ganges basin. In each month, the magnitudes of precipitation from retrospective (i.e., CHIRPS) and hindcast simulations (i.e., downscaled GEOS5-S2S-V1) are similar to each other due to the application of CDF matching (see section 2b). The inter-annual variation of the hindcast precipitation and retrospective precipitation both are higher in wet seasons than in dry seasons. The inter-annual variation of hindcast

precipitation, however, is smaller than that of the retrospective precipitation, especially in months with large lead time. This difference in inter-annual variability is the result of averaging across large forecast ensemble spread in later months.

Fig. 4b shows the climatology RZSM (i.e., average RZSM from the year 2000 to the year 2017) for May to January from retrospective and hindcast-RIC simulations. The climatologies of RZSM from retrospective and hindcast-RIC simulations have similar magnitude and seasonality. The magnitudes of inter-annual variability of each month, which are represented by inter-

annual standard deviation and are shown as error bars in 4b, are smaller in hindcast-RIC simulations. The reason for this magnitude difference is, again, the effect of averaging the large ensemble spread of hindcast-RIC simulations. Climatologically, as the monsoon season picks up in July, RZSM increases dramatically due to intense precipitation. The RZSM climatology peaks in August, indicating some delay between the precipitation peak and the annual maximum in soil water storage.

Fig. 4c shows the RZSM standardized anomaly and rainfall inter-annual correlations between retrospective and hindcast-RIC simulations for each month in the Ganges basin. Precipitation has positive correlations during the rainy seasons, but this prediction skill is never statistically significant. At the end of the monsoon (October), precipitation skill drops dramatically to near zero. In contrast, the correlation for RZSM (yellow line in Fig. 4b) fluctuates around the significant line (dashed line in Fig. 4b, R=0.484) as lead time increases. This fluctuation of RZSM skill within the monsoon season is closely driven by the

performance of the meteorological forecast of precipitation (green line in Fig. 4b). Although the skill in precipitation drops rapidly after the monsoon season, the RZSM forecast skill maintains around the significant line with a mild decline trend. This difference indicates that the low precipitation amount after monsoon seasons has little influence on the interannual variability of the RZSM. The high basin-scale RZSM correlation during July to December (yellow line in Fig. 4b) is a contrast to the relatively low pixel-scale RZSM in Ganges Basin (Fig. 3c-h). This difference reflects the fact that small spatial mismatch



between retrospective and hindcast-RIC simulations, which is likely due to the same mismatch of meteorological forcings, especially precipitation (see blue line in Fig. 4b and Fig. S4), can lead to disagreement at pixel scale that averages out at basin scale.

Fig. 4d shows the monthly RZSM standardized anomaly (see section 2e) from retrospective and hindcast-RIC simulations. The ensemble means of RZSM from hindcast-RIC simulations are shown in red dots, and the error bars represent the standard

deviation of the ensemble. The spread of the ensemble members is relatively small in May, which represents high confidence in the prediction from the forecast system. This confidence originates from the relatively low uncertainty of meteorological forcing variables in May. The spread increases quickly in June (i.e., 2-month lead time) and later lead months.

The timing of the Indus basin rainy season is similar to the Ganges, but the magnitude of precipitation is far less (Fig. 5). As a result, the average RZSM from all years is much lower in the Indus (Fig. 4b and Fig. 5b). Precipitation amount increases from

May to June, but high evapotranspiration (ET) in June causes a deficit in the water budget that causes a decrease of RZSM in June. Though climatology of precipitation is low from November to January, monthly RZSM climatology still increases during this period due to low ET. Hindcast-RIC simulations underestimated climatological RZSM in every month, and the underestimation grows in later dry months. This underestimation is, in part, due to the high bias of ET in the hindcast-RIC simulations, which is an issue of ongoing study. The precipitation correlations between hindcast-RIC and retrospective

simulations are low and statistically insignificant for all months, as the maximum correlation is 0.22 in May (red line in Fig. 5c). Though hindcast-RIC simulations show lower prediction skill in precipitation in the Indus than in the Ganges basin, the RZSM correlations in the Indus are still significant in the first three months (orange line in Fig. 5c). This relatively strong RZSM prediction skill is again a result of a drier climate. The relatively low amount of precipitation in the Indus basin allows the skill derived from initial conditions to persist longer.

In the Helmand basin, the magnitude of precipitation is smaller than in either the Ganges or Indus basin. More importantly, the seasonal precipitation cycle is different as well (Fig. 6a). The Ganges and Indus basins experience peak precipitation during the summertime South Asian monsoon, such that our May forecast initialization captures the onset of the rainy season. The Helmand basin receives precipitation in winter/spring, with the dry season setting in in May, just as our forecasts initialize (Fig. 6a). This allows us to see the behavior of a forecast that is nearly pure persistence: there is some precipitation in May,

for which hindcasts show skill, but for the following 5-6 forecast months, there is effectively no precipitation, and soil moisture shows a steady decline (Fig. 6b). This allows skill present in the initial conditions to persist, as seen in the high RZSM correlation between hindcast-RIC and retrospective simulations through November (Fig. 6c). This high correlation starts to drop from November but remains significant until December, even though precipitation has low prediction skills in November and December. While this result might not be indicative of seasonal forecast skill in the Helmand basin in general, since

forecasts initialized for the rainy season might be of broader relevance, the result does provide a useful example of how our forecast system behaves in basins initialized during the dry season.



## 3.2 Impact of initial conditions

The RZSM forecast skill is a product of the accuracy of both the forecast meteorological forcing variables and initial hydrological states. In this section, we examine the impact of initial conditions on prediction skills by comparing hindcast-RIC

simulations to our hindcast-CIC simulations (see section 2d for details). We focus on RZSM correlations with the retrospective simulation as our performance metric, since we are most concerned with the ability of quality forecast initialization to improve the simulation of interannual RZSM variability. Fig. 7 shows the contribution of initial conditions as the difference in correlation with retrospective RZSM between hindcast-RIC and hindcast-CIC simulations. In these maps, shades of red indicate areas and times where hindcast-RIC have improved skills relative to hindcast-CIC simulations. For lead-one forecasts

(May), we see a positive contribution of initial conditions to RZSM forecast skills primarily in India. For lead-two forecasts (June), the area with the largest correlation difference moves to the west of the domain, and the area with a significant correlation difference in southern India also shrinks. In later months, from July to December, the areas with significant differences of correlations continue to shrink because precipitation, along with other meteorological forcing variables, gradually reduces or even eliminates the impact of initial conditions. In portions of the west of the domain (Helmand basin),

where the precipitation is low from May to November, the difference in correlations remains significant until December.

Fig. 8 shows the number of lead months before the difference in RZSM forecast skill—measured as correlation with the retrospective simulation—from the two sets of hindcast simulations (hindcast-RIC and hindcast-CIC) drops to a negligible value (in this case, 0.01). Given that the only difference in the two sets of hindcast simulations is the initial conditions, Fig. 8 indicates the duration of the impact of the initial conditions on the performance of hindcast simulations. Blueish color in this

figure indicates areas with short memories of the initial condition. We note that our hydrological hindcasts are based on "offline" simulations in which surface conditions like soil moisture do not affect meteorological forcings. The influence of the initial conditions, then, is limited to direct impacts of water storage on the water balance, and does not include any potential land-atmosphere feedbacks. Short memories generally present in areas receiving considerable summer monsoon precipitation, such as the west coast and northeast parts of India, the southern slopes of the Himalayas, the west coast, and the southeast of

the Indochinese peninsula. The soil in these wet areas becomes saturated during the monsoon seasons due to intense precipitation. The memories of the initial conditions are thus weakened or eliminated. In contrast, having less precipitation, the yellow areas have longer memories of the initial conditions. This result is specific to our initialization month, and will differ with respect to the timing of the rainy season.

At basin scale, we see significant variability in the role that initial conditions play in RZSM forecasts. For the Ganges basin,

the correlation of RZSM between the retrospective simulations and the hindcast-CIC simulation is -0.04 in May (Fig. 4c green line) due to low precipitation correlation and inconsistent initial condition between the retrospective and hindcast-CIC simulations. A significant difference between hindcast-RIC RZSM (Fig. 4c orange line) and hindcast-CIC RZSM (Fig. 4c green line) in terms of correlation with retrospective RZSM in May suggests that an accurate initial condition is crucial to have meaningful RZSM prediction. This difference, however, becomes less in June and July. The RZSM predictions from hindcast-





RIC and hindcast-CIC simulations become almost identical after July as the memory of the initial condition is eliminated by the accumulating effects of meteorological forcing variables.

In the drier Indus basin, the difference in RZSM prediction skills between the hindcast-RIC and hindcast-CIC (Fig. 5c yellow and green lines) reduces over the first three months. This difference, however, is never eliminated. The difference between these two correlations becomes roughly constant from August to January. In January, nine months after initialization, the

RZSM from hindcast-RIC simulations still shows skillful prediction while the RZSM correlation between the hindcast-CIC and retrospective simulations becomes negative. This result indicates that in a drier basin, the impact of the initial conditions is reduced but can still positively contribute to the prediction skills of RZSM even nine months after the initialization date. In the Helmand basin (see the location of Helmand basin in Fig. 1), both hindcast-RIC and hindcast-CIC simulations have significant correlations with RZSM in retrospective simulations at lead-one month (May) due to high agreement between

hindcast and retrospective precipitation products, but the hindcast-RIC simulation is still significantly better than hindcast-CIC in this month. The predictability of RZSM in hindcast-CIC simulations brought by skillful precipitation prediction in May is quickly lost in June and drops to negative values from July to January, while the hindcast-RIC simulation has skillful RZSM predictions through December on account of the long memory of initial conditions through the protracted Helmand basin dry season.

**3.3 Comparison with satellite estimates**

The top-10cm surface soil moisture (SSM) from retrospective and hindcast-RIC simulations is also evaluated against the ESA-CCI SM products (see Section 2b for details). SSM values extracted from the ESA-CCI SM product, retrospective simulations, and hindcast-RIC simulations are pre-processed to a monthly time scale before comparison. The monthly ESA-CCI SM data are calculated by averaging all available daily data for that month. To make a fair comparison, we calculate retrospective and

hindcast-RIC monthly SSM data by first upscaling daily data to the same resolution as ESA-CCI SM ($0.25° \times 0.25°$) and then averaging to monthly time scale using only data from days when ESA-CCI SM daily data are also available.

Fig. 9 shows the inter-annual correlation map between retrospective monthly SSM and ESA-CCI SM (left) and the inter-annual correlations between hindcast-RIC monthly SSM and ESA-CCI SM (right) using data in May, July, and September between 2000-2017. The retrospective SSM in May has a high agreement with ESA-CCI SM in areas in south India, the Indochinese

Peninsula, and Northern Afghanistan. In July and September, the retrospective simulation captures more spatial patterns of ESA-CCI SM than in May in Pakistan and India, but less in Afghanistan and the Indochinese Peninsula.

Hindcast-RIC SSM generally has a lower correlation with ESA-CCI SM than retrospective SSM. In May, hindcast-RIC SSM has a significant positive correlation with ESA-CCI SM in the Indochinese Peninsula and Northern Afghanistan. In July and September, the forecast skill of SSM dramatically drops. Areas with significant SSM only appear sporadically in India, Nepal

and the Indochinese Peninsula.

Fig. 10 shows basin-scale comparisons with ESA-CCI SM, with precipitation seasonality included for context (Fig. 10a). At this scale we see high correlations between the retrospective simulation and ESA-CCI SM in most basins in most months:



correlations are significant in the Indus basin in all months; in the Helmand basin in all months but September; in the Brahmaputra basin in all months except for November; in the Ganges basins in all months except for November and December; and in the Mekong basin in all months except for August and September.

As expected, correlations are lower for the hindcast-RIC simulations. In May, there are significant skills in all basins but the Ganges basin, where the hindcast-RIC had also shown low skill relative to retrospective simulations (see Fig. 4). In June, skill drops off surprisingly quickly in the Indus and Helmand basins, both of which showed extended RZSM forecast skill when evaluated against the retrospective simulation (see Fig. 5 and Fig 6). This occurs despite the fact that ESA-CCI SM and the retrospective run have a relatively high correlation in these months. One reason that skill relative to ESA-CCI SM might drop off so quickly in these drier basins is that the memory of SSM, as opposed to RZSM, is short, especially in dry areas, and SSM can be highly sensitive to modest rainfall. This could make the ESA-CCI SM evaluation more sensitive to errors in forecast precipitation (see precipitation correlation in Fig 5 (c), 6 (c)) than the RZSM evaluation was, leading to a rapid loss of skill due to imperfect precipitation forecasts in the first few months of the simulations. From July onward the basin-scale hindcast-RIC evaluations relative to ESA-CCI SM are noisy; all basins drop to non-significant correlations in July and August, but there is a rebound to skillful prediction in the Mekong and Helmand in September due to the rebound of the precipitation forecast skill (see precipitation correlation in Fig. 6 (c) and A2 (c)). We also note that in the Helmand basin, the SSM correlation in October shows the opposite tendency from the precipitation correlation (Fig. 6c). The difference could be due to errors in ESA-CCI SM or CHIRPS, the influence of missing data in ESA-CCI SM, or noise in the forecasts. By November, the correlation between ESA-CCI and hindcast-RIC SSM drops to insignificant levels for all basins. It is worth noting that although on the basin scale we see significant SSM correlation in the Brahmaputra basin in September, spatial SSM correlations suggest lower forecast skills (Fig. 9). This difference in correlation suggests a spatial mismatch in the SSM predictions. A similar difference in correlation of spatial precipitation and basin-scale precipitation is also found in the Brahmaputra basin (Fig. 5(c) and A4).

**3.4 Case study of the 2015 South and Southeast Asia drought**

The 2015 El Niño event caused widespread drought in South to Southeast Asia. This drought had significant impacts on health, food security, and fire risks in more than nine countries in south-southeast Asia (Van Der Schrier et al., 2016;Qian et al., 2019). Fig. 11 shows the performance of our system in monitoring and forecasting the development of the 2015-2016 drought in the Ganges and Mekong basins. This hindcast is initialized on May 1st, 2015.

In the Ganges basin, the precipitation (Fig. 11a) and RZSM (Fig. 11e) values from the hindcast generally match the magnitudes of the retrospective simulations. There is some month-to-month discrepancy in precipitation hindcast relative to the CHIRPS record used in the retrospective simulation, but over the course of the monsoon season, these differences nearly average out— the hindcast only slightly overestimates precipitation (and thus underestimates the severity of the drought). This results in an RZSM forecast that is generally consistent with the observations, albeit somewhat less dry overall and noisier month-to-month, up until the final month of the forecast.


In the Mekong basin, in contrast, the forecast underestimates monsoon season precipitation (overestimates drought severity) (Fig. 11b, d). This results in a hindcast in which drought is both more severe and more persistent than observed (Fig. 11 f, h). Thus, the direction of the drought is captured in this basin, but in application the hindcast might overestimate the predicted impacts of this 2015 drought.

**3.5 Application to drought**

Applying a hydrological monitoring and forecast system to drought applications requires that the ensemble simulation output be converted to meaningful and interpretable drought indicators. Here we use the Mekong and the Helmand basins as examples to illustrate one method for doing so. This process starts with the monthly RZSM values from both retrospective and hindcast-RIC simulations. In each month, the percentile of the monthly RZSM values is calculated on a gridded basis, based on a normal distribution fitted from the 18-year data record at each grid cell. The severity of drought at each grid cell is then categorized

based on the same drought categories used in the United States Drought Monitor (Svoboda et al., 2002): the location is classified as exceptional drought (D4), extreme drought (D3), severe drought (D2), moderate drought (D1), or abnormally dry (D0) using RZSM percentile thresholds of 2%, 5%, 10%, 20%, and 30%, respectively.

Figs. 12 and 13 show the fractional area of each drought category for the retrospective simulation and the hindcast-RIC simulations, for all lead times, in the Mekong and the Helmand basins, respectively. In the Mekong basin, the hindcast

simulations capture the major drought events reasonably well in May (1-month lead time), though they overestimate the drought areas in 2005 and underestimate them in 2010 (Fig. 12a). From June to August (2-month lead time to 4-month lead time), the hindcasts simulations predict significantly larger drought areas in 2005 and 2016 compared to retrospective simulations, while hindcast-RIC and retrospective simulations estimate relatively similar drought areas in other drought years (Fig. 12b-d). After August, the drought areas become poorly estimated in hindcast-RIC simulations compared to the

retrospective simulations (Fig. 12e-i) due to unskillful hindcast-RIC RZSM estimates in the Mekong basin (Fig. S2c). If we focus on the 2015 drought event, the hindcast-RIC and retrospective simulations agree well in the first three months (Fig. 12a-c). Starting from August, the hindcast-RIC simulations estimate larger drought areas than the retrospective simulations, especially in the "exceptional drought" category (Fig. 12d-i). This overestimation of the drought area is consistent with the lower prediction of the RZSM standardized anomaly in Fig. 11. The progress of a specific drought event can also be tracked

by comparing drought categories cross different months (see red rectangle in Fig. 12 and Fig. S5 for 2015 drought).

Consistent with a high correlation of RZSM between retrospective and hindcast simulations (Fig. 6b), the fractional drought area in the Helmand basin agrees well for the first seven months (Fig. 13a-g). From the year 2000 to 2004, the Helmand basin experienced severe drought conditions. In 2005, above-average rainfall ended this prolonged period of drought conditions. In 2016, the drought condition is relatively stable from May to October and then becomes more severe in November and

December according to retrospective and hindcast-RIC simulations. In January, however, retrospective simulations report a less severe drought than December, while hindcast-RIC simulations report a more severe drought. It is worth noting that an intensification of drought from one month to the next does not necessarily mean that the soil moisture is drier than the previous





month. Each month's drought indices are calculated relative to the distribution of historical soil moisture conditions in that month.

**4. Conclusion**

In this study, we present a high-resolution soil moisture monitoring system and sub-seasonal to seasonal forecasting system for a South and Southeast Asia region, SAHFS-S2S. SAHFS-S2S consists of a physically-based land surface model, analysis and observation-based meteorological forcing datasets, and downscaled dynamically-based meteorological forecasts. We compare 18 hindcast-RIC simulations, each of which is initialized on May 1st in a year from 2000 to 2017, with corresponding
retrospective simulations. The comparisons show that the RZSM in hindcast-RIC simulations have considerable skill for the first two months, especially in the western part of the study domain, the Indochinese Peninsula, and southern India. The hindcast-RIC simulations continue to have high skills in the western part of the research domain—which are the driest areas in the domain—for another five months while showing generally low skill in other regions. Results presented here only capture system skill for forecasts initialized in May, but diverse seasonality across the study domain allows us to examine forecast
performance for both wet and dry season initialization dates.

To study the impact of the hydrological initial conditions on forecast skill, we designed a set of control hindcast simulations, which are initialized with climatological hydrological conditions (hindcast-CIC). In May (1-month lead time), the hindcast-RIC simulations outperform hindcast-CIC simulations in most parts of the domain except for the Indochinese Peninsula, where heavy precipitation quickly eliminates the memory of the initial conditions. The difference between the hindcast-RIC and
hindcast-CIC simulations decreases as lead month increases, and the accumulated influence of meteorological forcing gradually overwhelms the impact of the initial condition. In the Indochinese Peninsula, India, and surrounding areas, where precipitation is relatively intense in summer seasons, the influence of initial condition on the forecast skill is eliminated after two to three months. The correlation of precipitation dominates the prediction skills of RZSM when such a considerable precipitation influence emerges. When precipitation is low, however, the prediction skill of RZSM depends on the RZSM
prediction in the previous month, with relatively less influence from other meteorological forcing variables. This pattern becomes particularly important in regions where the initialization dates (i.e., May 1st) of hindcast-RIC simulations are around the beginning of day seasons. For example, in the Helmand basin, where the precipitation is mainly from mountain precipitation during the winter and early spring months (see section 2.1), the accurate initial conditions help the prediction skills remain statistically significant for eight months.

We also compared the surface soil moisture (SSM) from retrospective simulations and hindcast-RIC simulations with the ESA-CCI SM data product in the five major river basins in the study domain. The comparison shows that the retrospective simulations capture interannual variability in most of the months within the five basins. The SSM hindcasts in four out of five major basins (all but the Ganges) generally have a high correlation with ESA-CCI SM data in the first one or two months. The





correlation then decreases for another one or two months, then increases again and reaches a second maximum after the peak

monsoon rains (for which precipitation forecast skill is limited) before decreasing for the rest of the forecast period.

The prediction skill of the forecasting system of SAHFS-S2S depends on the land surface model, initial conditions, and meteorological forecasts. In this study, an accurate initial condition has been shown to have a positive contribution for prediction skills over much of the simulation domain, and particularly in dry areas and seasons. Future effort should be made to improve the accuracy if initial conditions estimated by the land surface model. This could include (1) assimilating ground

or satellite-based observations into the land surface model (Getirana et al., 2020a;Getirana et al., 2020c;Wanders et al., 2013), and (2) better representation of anthropogenic influences, for example, irrigation (Nie et al., 2019) and reservoirs (Wanders and Wada, 2015;Getirana et al., 2020b), in the land surface model / hydrological model. Errors in the meteorological forecast are also a clear limitation on the forecast skill. Improved dynamically-based S2S meteorological forecast systems (Pegion et al., 2019), advanced statistical-dynamical forecast methods (Madadgar et al., 2016;Shukla et al., 2014), and improved bias

correction and downscaling methods (Rodrigues et al., 2018) are all areas of significant research effort. The results presented here show that current capabilities offer meaningful skill over shorter time horizons for much of the domain, and also that performance can be improved as each component of the forecast system improves.

Due to the difficulty of acquiring reliable, long-term streamflow observation, it is difficult to evaluate the streamflow monitoring and forecasting in this study domain. However, Yang et al. (2011) evaluated Noah-MP forced by Global Land

Data Assimilation System (GLDAS) in the Mekong and Ganges basins in this study domain and found that Noah-MP captured the seasonality of the streamflow in both basins and magnitude in the Mekong basin but underestimated the magnitude in the Ganges basin. Ghatak et al. (2018) found that Noah LSM forced by GDAS and CHIRPS captured the timing of a one-year flood event in the Indus basin but underestimated the magnitude and captured both seasonal cycles and magnitude of streamflow in the Kosi basin within the Ganges basin. The streamflow from the SAHFS-S2S monitoring system has been

found to have similar results to those in Ghatak et al. (2018).

The SAHFS-S2S has been operationally implemented at the International Centre for Integrated Mountain Development (ICIMOD). The monitoring system updates about every ten days and the forecast system launches a 9-month hydrological forecast at the beginning of each month. The outputs of SAHFS-S2S have been integrated to ICIMOD's Regional Drought Monitoring and Outlook System (RDMOS), which focuses on crop and drought conditions mainly within countries in the

Hindu Kush Himalaya regions. This integrated system is designed to support local decision makers and government agencies concerned with food security and drought preparation.

**Data availability**

Operational SAHFS-S2S are available through ICIMOD data portals (http://tethys.icimod.org/apps/regionaldrought/current/). The hindcast simulation output used in this evaluation study will be made available through the Johns Hopkins University

Data Archive (https://archive.data.jhu.edu/) once the final version of this discussion paper is published.





**Author contribution**

BZ and SK formulated the idea of the monitoring and forecasting system. YZ, RA, BZ, and KA designed the workflow of the monitoring and forecasting system. YZ implemented monitoring and forecasting system and performed the retrospective and hindcast simulations with help from SK, BZ, RA and KA. YZ and BZ analysed the performance of the monitoring and
forecasting system. YZ, MM, FQ and KS oversighted, managed and implemented the system in operational mode at ICIMOD. YZ prepared the first draft of the manuscript with contributions from all co-authors.

**Competing interests**

The authors declare that they have no conflict of interest.

**Disclaimer**

The views and interpretations in this paper are those of the authors and are not necessarily attributable to JHU, ICIMOD, NASA or USAID.

**Acknowledgements**

The authors would like to thank Hiroko Kato Beaudoing and Jossy Jacob from NASA/GSFC for processing and maintaining GDAS dataset over the years, Hamada Badr and NASA LIS team for the help when setting up LIS, Niklas Troxel and Umesh
Upadhyaya for software/hardware technical support.

**Financial support**

This research has been supported by NASA Applied Sciences Program SERVIR award NNX16AN38G.





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




**Figure 1: (a) The extent of simulation area, research domain and five major river basins in South and Southeast Asia; Seasonal precipitation climatology for 2000-2018 (b) December, January, and February; (c) March, April, and May; (d) June, July and August; (e) September, October and November. The seasonal precipitation climatology is estimated from Climate Hazards Center InfraRed Precipitation with Station (CHIRPS) data.**




**Figure 2:** A schematic workflow of the South and Southeast Asia monitoring and forecasting system. Retrospective simulations (open-loop), hindcasts with "Real" initial condition (hindcast-RIC) simulations, and hindcasts with climatological initial condition (hindcast-CIC) simulations are designed to evaluate the monitoring and forecasting system.


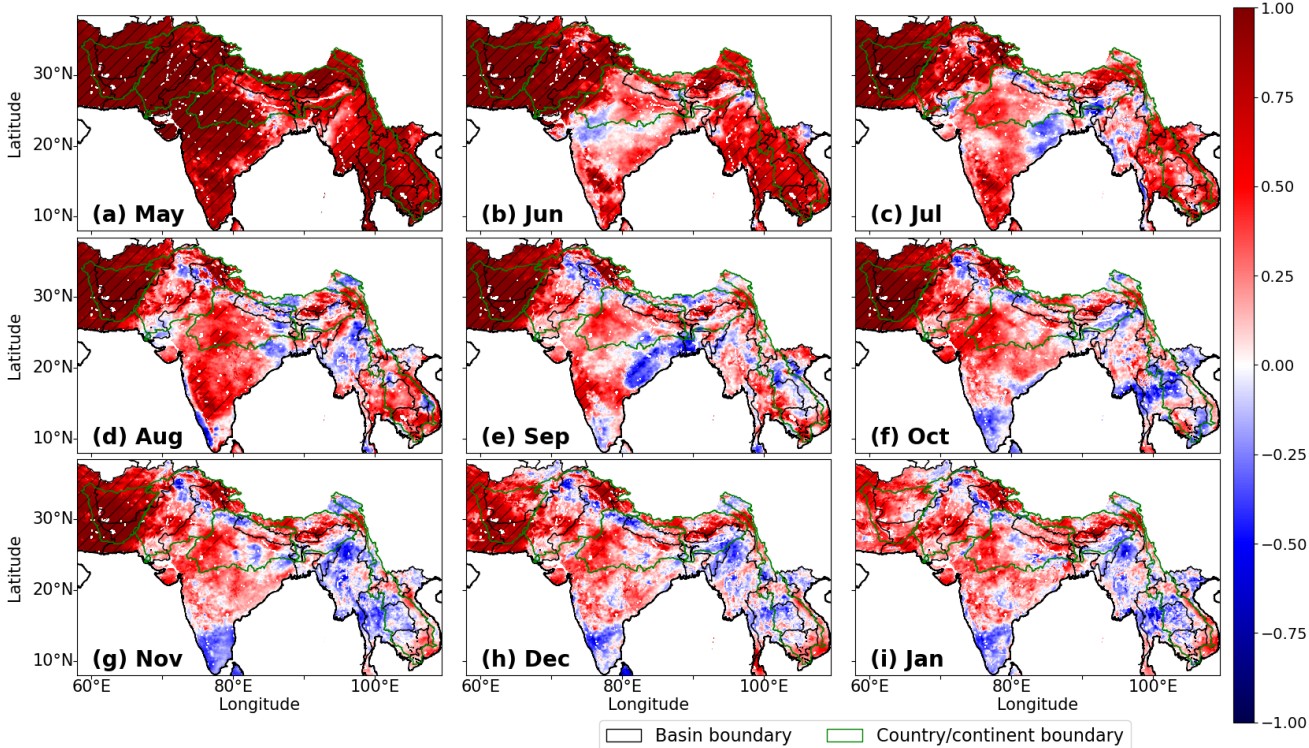

**Figure 3: Interannual correlation of RZSM between retrospective simulations and hindcasting simulations which is computed using RZSM data from year 2000 to 2017 in months May (1-month lead time) to January (9-month lead time) (subplot (a) to subplot (i)). The hatches denote the areas with statistically significant correlation at 0.95 confidence level.**






**Figure 4: Comparison between retrospective simulations and hindcast simulations in the Ganges basin of (a) monthly time series of precipitation, (b) monthly climatology of root zone soil moisture (RZSM), (c) inter-annual correlations of basin-averaged RZSM standardized anomaly and precipitation, and (d) monthly standardized anomaly of RZSM (the red error bars represent for the standard deviation of separate hindcast ensemble members). Please note that the time axis for monthly time series are rearranged so that the data for the same month is grouped.**






**Figure 5: Same as Fig. 4, but for the Indus basin.**

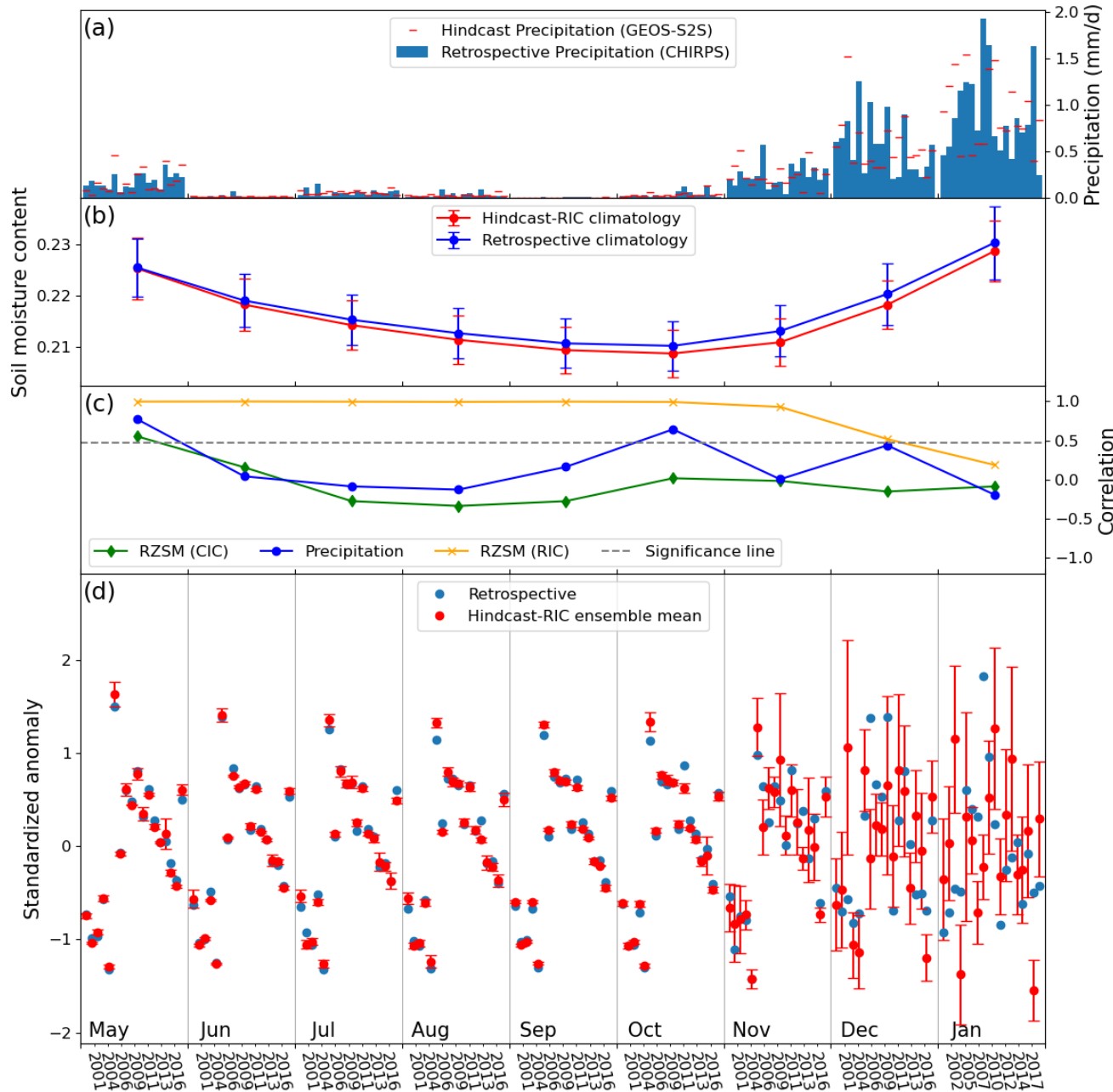

**Figure 6: Same as Fig. 4, but for the Helmand basin.**





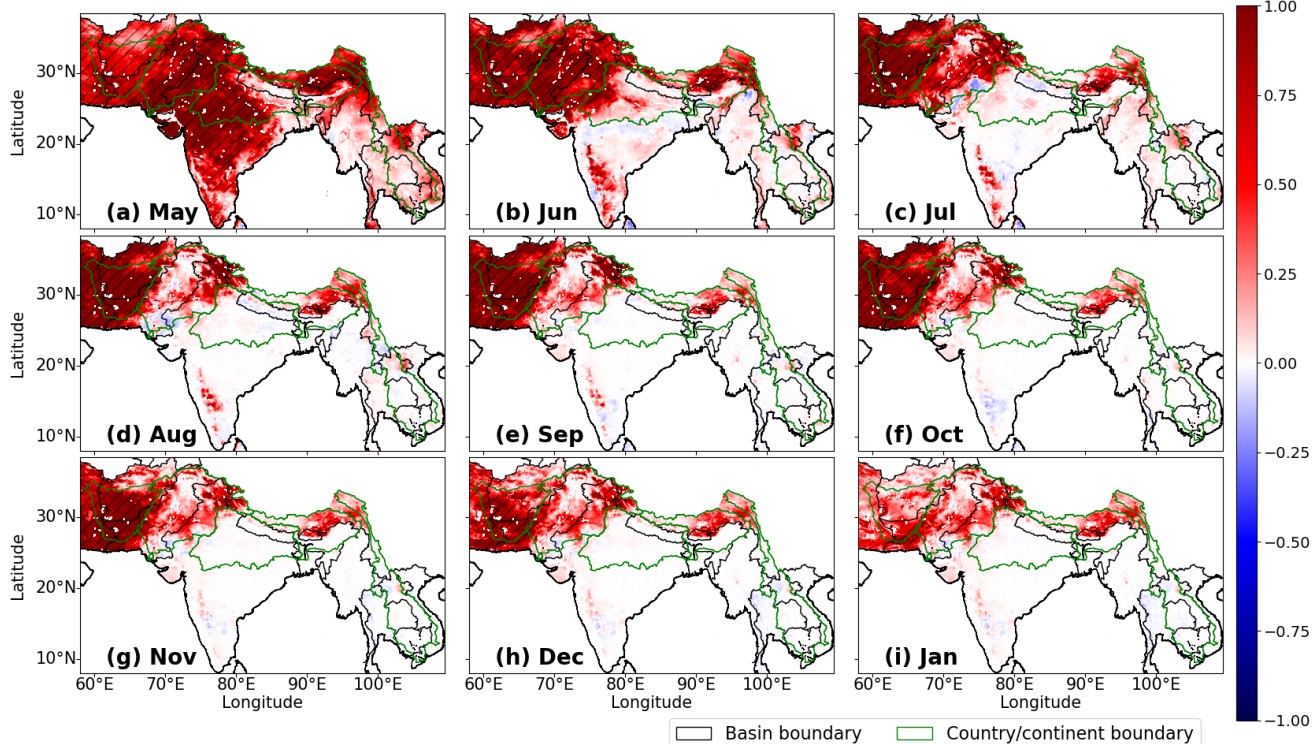


**Figure 7: Interannual correlation difference of root zone soil moisture (1 m) between hindcast simulations using real initial condition (RIC) and climatological initial condition (CIC) against the retrospective simulations. The difference is computed as the correlation of RZSM between hindcast simulations and retrospective simulations minus the correlation between hindcast simulations and retrospective simulations. The hatched areas denote the statistically significant correlation difference at 0.95 confidence level.**


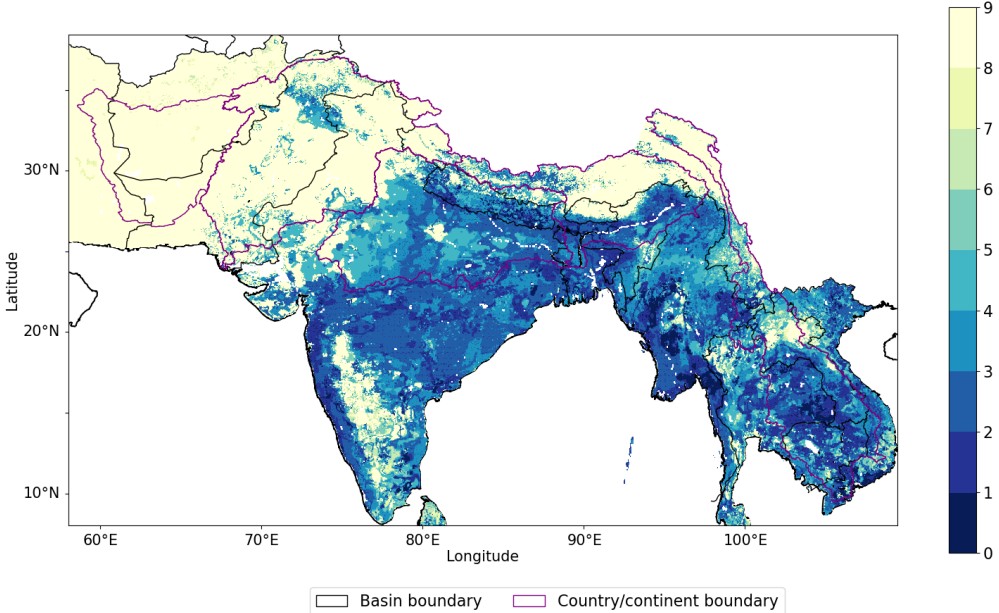

**Figure 8: The smallest lead month when the forecast skills of root zone soil moisture (1m) between hindcast simulations using real initial condition and climatological initial condition (CIC) against retrospective simulations is less than 0.01.**


**Figure 9: Surface soil moisture (top 10cm) correlation between European Space Agency Climate Change Initiative (ESA CCI) soil moisture product and (a) retrospective simulations, and (b) forecasting simulations in May, July, and September. The hatched areas denote the statistically significant correlation at 0.95 confidence level.**


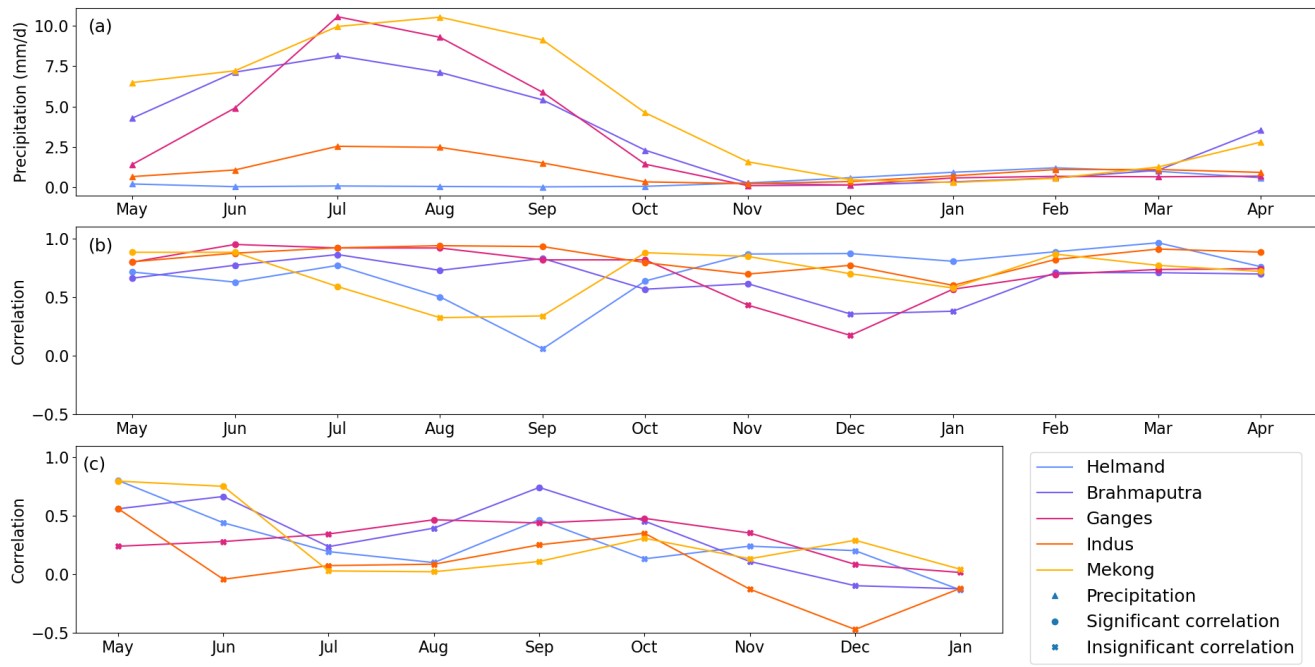

**Figure 10: Comparison of (a) monthly retrospective precipitation climatology, (b) interannual correlation between monthly ESA-CCI SM product and monthly retrospective surface soil moisture (top 10cm) and (c) interannual correlation between monthly ESA-CCI SM product and monthly forecasting SSM in five major river basins in South and Southeast Asia. Please note that the SSM data for a basin in a year is flagged as missing data if more than 50% of data points in a basin are missing in ESA-CCI SM monthly data. The criterion of significance of correlation are then different due to different sample sizes.**



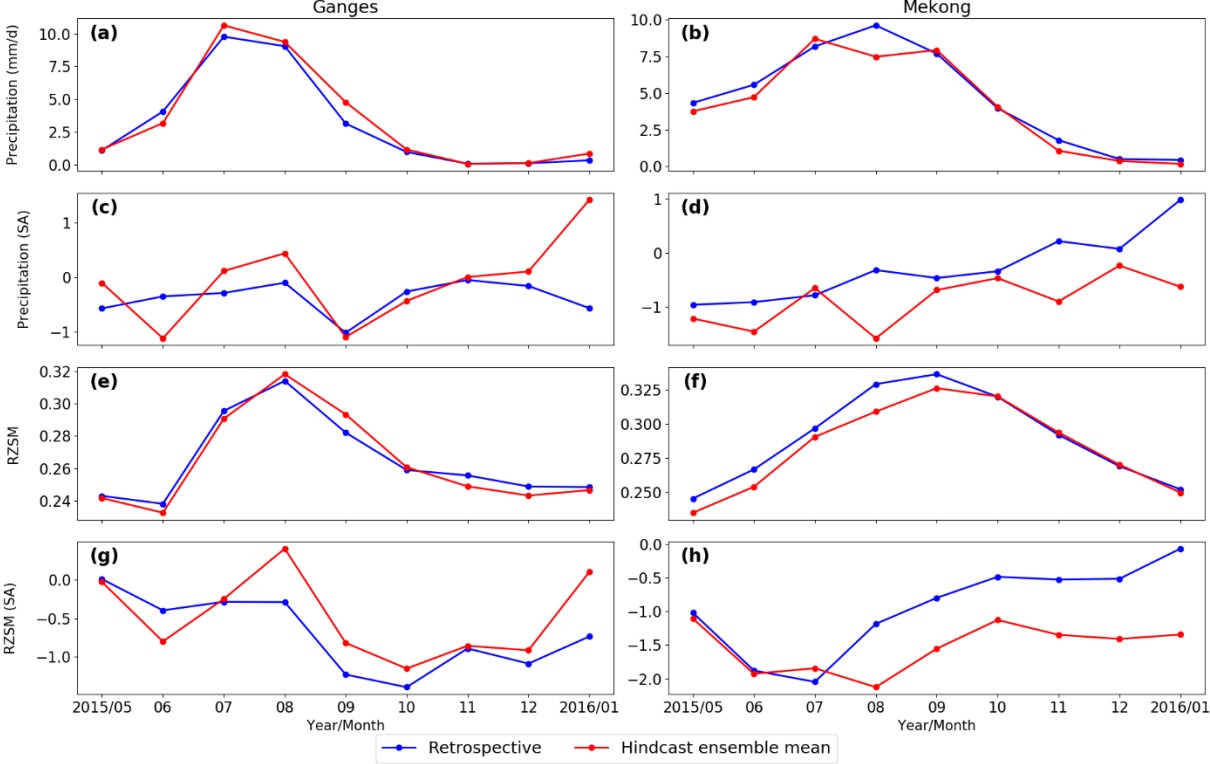


**Figure 11: Comparison of monitoring and forecasting system on 2015-2016 drought event in Ganges basin (left panels) and Mekong basin (right panels) in terms of precipitation (a, b), precipitation standardized anomaly (c, d), root zone soil moisture (RZSM) (e, f); and RZSM standardized anomaly (g, h).**






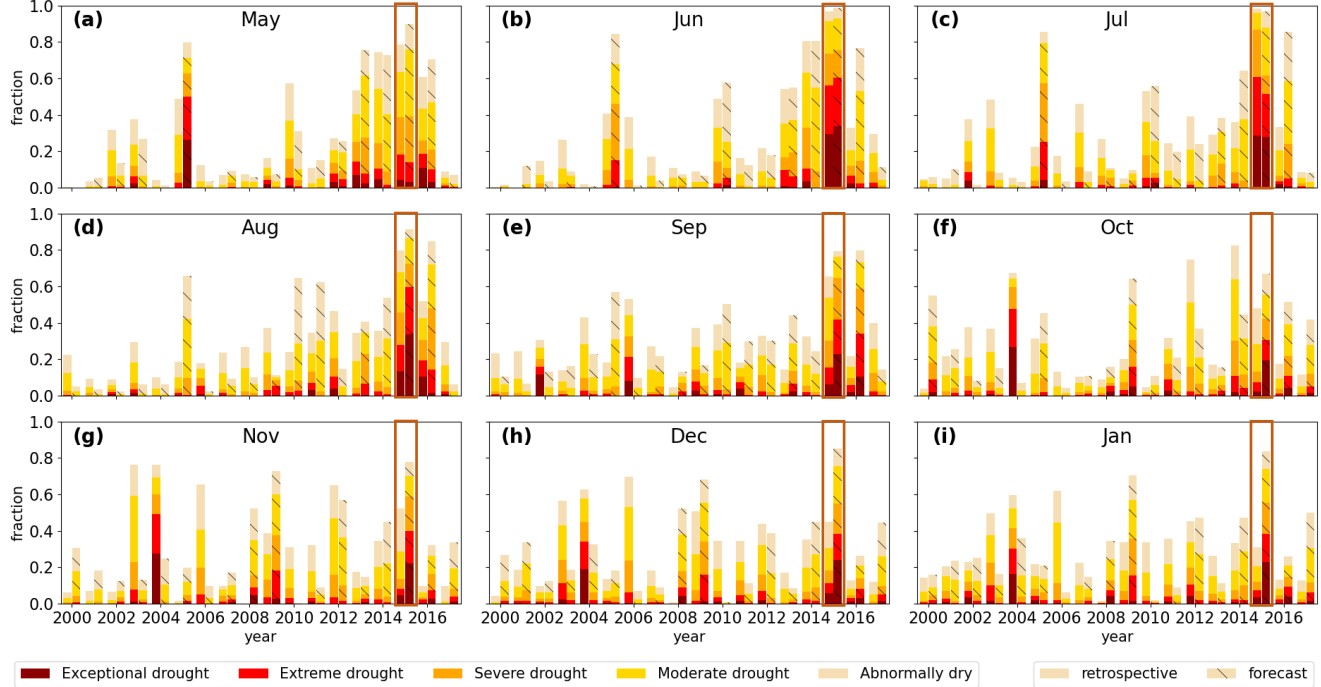

**Figure 12: Fractions of areas of different drought categories in Mekong basin calculated using RZSM data from retrospective and hindcast-RIC simulations from (a) May to (i) January.**






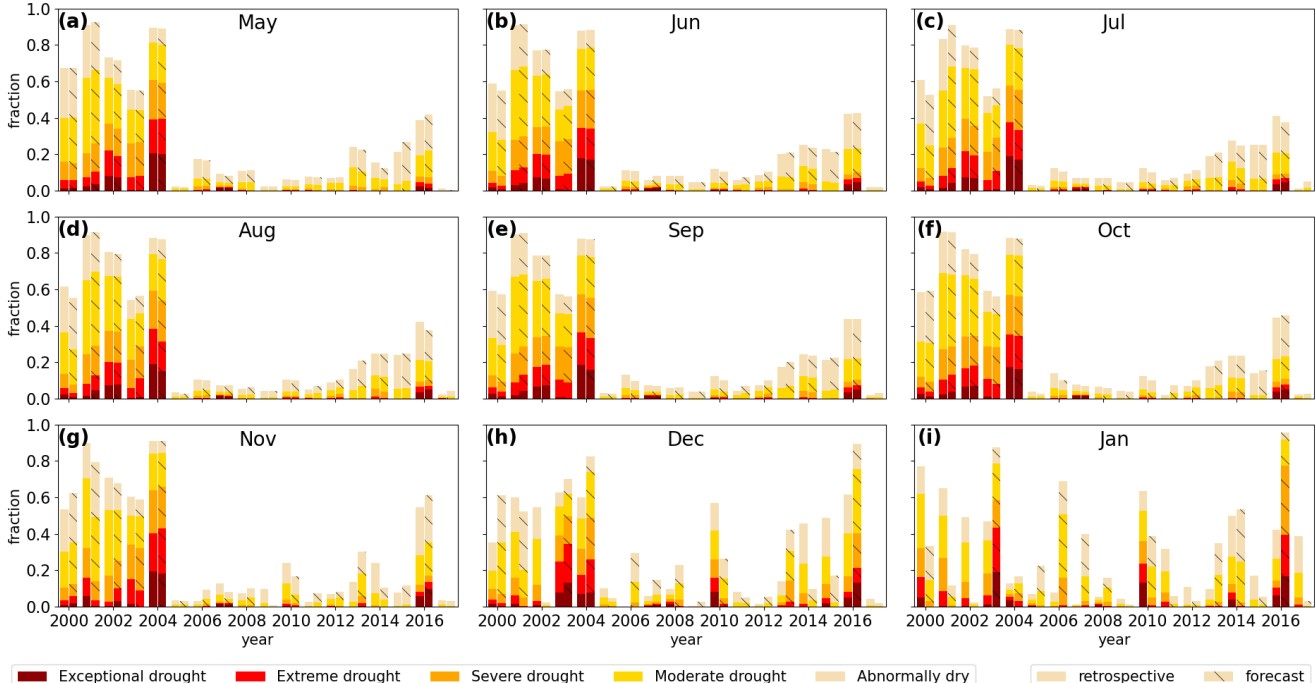

**Figure 13: Same as Fig. 12 but for the Helmand basin.**