# Peer review of "Developing a hydrological monitoring and sub-seasonal to seasonal forecasting system for South and Southeast Asian river basins"

_Hydrology and Earth System Sciences, 2020_

## Referee Comment (RC1) · Anonymous Referee #1 · 21 Aug 2020

This study develops a sub-seasonal to seasonal drought forecasting system for South and Southeast Asian river basin based on climate forecast and hydrological modeling. The performance of the forecasting initialized at May 1 for an 18-year period is evaluated against satellite retrievals and case studies. Overall, this is a well-crafted study with clear structures. I have some minor comments as follows.

Line 46-49: Repeated statements

Line 167: "2000-2018" and later in line 183 "from 2000 to 2017". Please double check.

Line 275: For forecast results of multiple members, the ensemble mean is based on standardized values of each member? Please clarify

[Figure]

Lines 286-293: It seems the correlation is particularly high in northwestern regions of the study area even after 3-6 months. Could you please explain this?

Section 3.4 "Case study of the 2015 South and Southeast Asia drought" and section "3.5 Application to drought". The "case study of 2015 drought" is not an "application to drought"? Please think about this and revise them accordingly if needed.

Line 472-474: This overestimation of drought area from August is due to the low performance of precipitation forecast during the monsoon season? Please clarify this overestimation.

---

## Referee Comment (RC2) · Anonymous Referee #2 · 24 Aug 2020

The article entitled "Developing a hydrological monitoring and sub-seasonal to seasonal forecasting system for South and Southeast Asian river basins" develops a sub-seasonal to seasonal hydrological forecasting system for South and Southeast Asia (SAHFS-S2S). The system applies the NoahMP land surface model, driven by CHIRPS for monitoring and driven by GEOS-S2S for forecasting. The system shows skillful predictions of root zone soil moisture one or two months in advance when initialized in rainy seasons and up to 8 months when initialized in dry seasons, due to the contribution from initial conditions. The results could provide end-users with water resources information to help manage local drought risks. It is an interesting study, and the paper is easy to follow. However, the conclusion of "the impact of initial conditions on

forecast skill depends on initialized dry/wet seasons" has been presented in previous researches. What's innovation in this paper except for the different study regions?

Refer to Yuan, X., F. Ma, L. Wang, et al., 2016: An experimental seasonal hydrological forecasting system over the Yellow River basin-Part 1: Understanding the role of initial hydrological conditions. Hydrology and Earth System Sciences, 20, 2437–2451. Luo, L., Sheffield, J., and Wood, E. F.: Towards a global drought monitoring and forecasting predictability, NWS Science & Technology Infusion Climate Bulletin, 2008. Shukla, S. and Lettenmaier, D. P.: Seasonal hydrologic prediction in the United States: understanding the role of initial hydrologic conditions and seasonal climate forecast skill, Hydrol. Earth Syst. Sci., 15, 3529–3538 Ma, F., Luo, L., Ye, A., et al. Seasonal drought predictability and forecast skill in the semi-arid endorheic Heihe River basin in northwestern China, Hydrol. Earth Syst. Sci., 22, 5697–5709, 2018.

The GEOS-S2S forecasts have been downscaled and bias-corrected before driving the hydrological model. Please added some more details of the downscaling algorithm and the performance before and after bias-corrected.

The impact of initial conditions is analyzed using the difference between hindcast-RIC and hindcast-CIC simulations. While many studies have analyzed the role of initial conditions by two experiments: Ensemble Streamflow Prediction (ESP) and reverse-ESP (revESP). Some discussion regarding the difference between them may be added.

Minor comments Line 40: The meaning of the sentence "The forecast period . . . in recent years" is not well understood. Line 44: What is "land component". Line 46: The sentence "The influence of initial hydrological . . ." is repeated. The depth of root zone soil moisture is 1 m? Some figures, such as Fig. S1, S3, S6, S7, in supplement information are not mentioned in the paper. The ESA-CCI SM derived from remote sensing observation has many missing data, so how to deal with the missing data? In data section, many datasets are used. Here, I suggest that a data table, including data sources, detailed information and variables used, should be listed for easily read. Hu-

man activities, such as irrigation, has great influence on soil moisture, how to consider irrigation in the NoahMP land surface model? Line 330: Please add a figure of monthly evapotranspiration (ET) for easily explaining. Line 780: Figure 7: The difference is computed as correlation between RIC and CIC? In the Case Study of the 2015 South and Southeast Asia section, I suggest a spatial distribution map showing the comparation of drought conditions between retrospective simulation and hindcast ensemble mean may be added. Figure 9: the tittle of right subplots should be (b), (d), (f).

---

## Author Comment (AC1) · 20 Oct 2020

We thank the referee for the efforts and constructive suggestions. We clarify the year range of the simulations, revise the section structures, and add explanations of overestimation of drought area in August based on your suggestions. The detailed responses are listed below.

Line 46-49: Repeated statements Response: We deleted one of the repeated statements in the revised manuscript.

Line 167: "2000-2018" and later in line 183 "from 2000 to 2017". Please double check.

Response: The "2000-2018" in line 167 is referring to the year range of spin-up simulation. We then use the restart file of Dec 31st, 2018 to initiate the retrospective run, which starts from 2000 and extends to 2017, therefore, in line 183, "the year 2000 to 2017" is referring to the year range for the retrospective simulation. We now modify the text in line 183 to avoid ambiguity.

Line 275: For forecast results of multiple members, the ensemble mean is based on standardized values of each member? Please clarify Response: Regarding the ensemble mean of the forecast results, we average all the ensemble members first and then standardized the averaged time series. We now clarify in line 275 as "For simplicity, the results presented in the following sections are only based on the standardized ensemble mean for forecast simulation related analyses, for which the standardization is applied after the calculation of ensemble mean".

Line 286-293: It seems the correlation is particularly high in northwestern regions of the study area even after 3-6 months. Could you please explain this? Response: The same initial conditions are used in hindcast-RIC and retrospective simulation. The northwestern region has relatively low precipitation and thus this region has a long hydrological memory which is drawn from the initial conditions. This phenomenon is further explained in detail in section 3.2. We now add this explanation to line 290.

Section 3.4 "Case study of the 2015 South and Southeast Asia drought" and section "3.5 Application to drought". The "case study of 2015 drought" is not an "application to drought"? Please think about this and revise them accordingly if needed. Response: We designed these two sections as we meant to use section 3.4 to show the capability of the model to capture a historical drought event and to use section 3.5 to illustrate the potential to use soil moisture-based drought indicators in general. We now modified the name of section 3.5 to "Application of drought indicator" in the revised manuscript.

Line 472-474: This overestimation of drought area from August is due to the low performance of precipitation forecast during the monsoon season? Please clarify this

overestimation. Response: Yes, and the overestimation of drought area for August is mainly attributed to the lower precipitation prediction from the downscaled GEOS-S2S-1, which is shown in Figure S1. We now add this explanation in line 474 in the revised manuscript.

---

## Author Comment (AC2) · 20 Oct 2020

We appreciate the efforts, constructive suggestions and the positive feedbacks by the referee. Following your suggestions, we enhanced our descriptions of the downscaling / bias-correction methods, and also add discussions related to ensemble streamflow prediction (ESP) / reverse ESP. The detailed responses are listed below.

1. The article entitled "Developing a hydrological monitoring and sub-seasonal to seasonal forecasting system for South and Southeast Asian river basins" develops a sub-seasonal to seasonal hydrological forecasting system for South and Southeast Asia (SAHFS-S2S). The system applies the NoahMP land surface model, driven by CHIRPS

for monitoring and driven by GEOS-S2S for forecasting. The system shows skillful predictions of root zone soil moisture one or two months in advance when initialized in rainy seasons and up to 8 months when initialized in dry seasons, due to the contribution from initial conditions. The results could provide end-users with water resources information to help manage local drought risks. It is an interesting study, and the paper is easy to follow. However, the conclusion of "the impact of initial conditions on forecast skill depends on initialized dry/wet seasons" has been presented in previous researches. What's innovation in this paper except for the different study regions? Refer to Yuan, X., F. Ma, L. Wang, et al., 2016: An experimental seasonal hydrological forecasting system over the Yellow River basin-Part 1: Understanding the role of initial hydrological conditions. Hydrology and Earth System Sciences, 20, 2437–2451. Luo, L., Sheffield, J., and Wood, E. F.: Towards a global drought monitoring and forecasting predictability, NWS Science & Technology Infusion Climate Bulletin, 2008. Shukla, S. and Lettenmaier, D. P.: Seasonal hydrologic prediction in the United States: understanding the role of initial hydrologic conditions and seasonal climate forecast skill, Hydrol. Earth Syst. Sci., 15, 3529–3538 Ma, F., Luo, L., Ye, A., et al. Seasonal drought predictability and forecast skill in the semi-arid endorheic Heihe River basin in northwestern China, Hydrol. Earth Syst. Sci., 22, 5697–5709, 2018.

Response: We acknowledge that the impact of initial conditions on forecasting skills are widely studied using different methods and applied in different regions, while the primary focus of our manuscripts is to establish a workflow of an operational sub-seasonal to seasonal hydrological forecast system for South Asia. Our study differs from the literature that the reviewer mentioned in the following aspects:

1) Our study domain – South Asia – is a challenging domain of interest to establish reliable S2S hydrological forecasts due to highly localized weather variability, complex hydrology, and active human water management. In our work, major efforts have been made to optimize the model settings, input parameters, forcing inputs, and the initializing dates to provide the best representation of the hydrological states and fluxes for

the domain, and establish a drought monitoring and forecasting system for the local stakeholders. In this regard, the most important unique contribution of the paper is the development and presentation of an operational S2S hydrological forecast system in a large and challenging region. The presentation of initial condition results for this region does echo previous literature, as the reviewer notes, and we cite that literature as appropriate. Given the diversity of climate zones and landscapes in our study domain, we feel it is important to understand and to quantify this initial condition sensitivity for the presented S2S forecast system, even if the topic has been addressed elsewhere in other studies. For example, recognizing the relatively long soil moisture memory for forecasts initialized in the dry season in the drought-prone and food insecure Helmand Basin has important implications for the potential to provide early food insecurity warnings. 2) Our study has a different consideration of the met-forcing fields. Compared with the above-mentioned studies that using monthly CFS, NMME-phase1, and DEMETER (Yuan et al., 2016;Shukla and Lettenmaier, 2011;Luo et al., 2008;Ma et al., 2018), the meteorological forcings of our study are drawn from GEOS5-S2S-V1 forecasts at daily resolution and downscaled based on GDAS+CHIRPS. Utilizing the information of day-to-day variation could potentially help improve the forecasting skills, which may alter the importance of initial condition. 3) In the above-mentioned manuscripts, the impact of the initial conditions is evaluated by comparing ESP and revESP. These experiments show the relative importance of perfect initial condition and perfect meteorological forcings without emphasis on hydrological forecast driven by meteorological forcings. The SAHFS compares two sets of hindcast simulations, hindcast with real initial conditions (hindcast-RIC) and hindcast with climatological initial conditions (hindcast-CIC). This comparison shows the relative importance of initial conditions in an actual forecast system driven by downscaled meteorological forecasts. 4) In our next step, the modeling platform that we built for this forecasting system, will be implemented with data assimilation techniques (Nie et al., 2018;Getirana et al., 2020) and is under development of including human water management such as irrigation (Nie et al., 2019). The inclusion of these parts may help to generate a better initial

condition, thus improving the forecasting skill.

2. The GEOS-S2S forecasts have been downscaled and bias-corrected before driving the hydrological model. Please added some more details of the downscaling algorithm and the performance before and after bias-corrected.

Response: We replace line 191-209 with the following text:

"In the forecasting system, the same meteorological forcing variables as in the monitoring system are extracted from GEOS-S2S data products (see details of GEOS-S2S in section 2b). Due to the coarse resolution and inevitable bias of global climate model outputs, these forcing variables from GEOS-S2S-V1 are downscaled and bias-corrected to corresponding monitoring system forcing variables (i.e., precipitation from CHIRPS and other variables from GDAS) using a Generalized Analog and Regression Downscaling (GARD) algorithm (https://github.com/NCAR/GARD; Gutmann et al. (2020)). This downscaling algorithm takes a training dataset, prediction dataset, and observation dataset as inputs. The observation dataset contains records of variables (dependent variables) with targeted fine spatial resolution, for example, the precipitation from the CHIRPS dataset. The training dataset includes records of coarse spatial resolution variables (independent variables), which have the same time resolution as the dependent variables, for example, hindcasted GEOS-S2S-V1 precipitation. The prediction dataset contains the records of the same coarse-resolution variables as the training dataset but acquired in the forecast period—for example, new GEOS-S2S-V1 forecast precipitation. The GARD algorithm downscales the prediction dataset to the resolution of the observation dataset with an analog-regression approach. This analog-regression approach takes two steps to downscale one variable at one-time step in one fine-resolution grid. In the analog step, the algorithm selects a user-defined number of training records whose values are the closest to the prediction records from the coarse-resolution grid cell that includes the fine-resolution grid cell. In the regression step, we then regress the selected training records on the observation records from corresponding time steps in the fine-resolution grid cell. The prediction record is fed to this trained

linear regression model to calculate the downscaled values of the variable at the targeted fine spatial resolution. In this application, we apply GARD to downscale each variable as a function of the same variable in the training datasets (e.g., precipitation to predict precipitation). GARD has the capability to use multiple predictor variables to improve downscaling accuracy, but the influence of the different combinations of independent variables is beyond the scope of this paper."

To address the evaluation of the downscaling method, we add Fig. S1-5 to supplements and the following text, beginning at line 229:

"The performance of the downscaling method is evaluated at the monthly time scale in the five major river basins in South Asia using root mean squared error (RMSE) (Fig. S1-5). The RMSE is first calculated for GEOS before and after downscaling against the retrospective forcing (i.e. the combination of GDAS and CHIRPS) and then normalized by the range of the retrospective forcing.

Overall, the RMSE of air temperature, surface pressure, and relative humidity is greatly reduced after downscaling. In addition to the GARD algorithm, the applied CDF matching has further reduced the RMSE for precipitation. For other fields, the impacts of downscaling differ across basins. For instance, downscaling leads to reduced RMSE of wind speed for the Ganges basin while its impact on shortwave and longwave radiation is marginal."

3. The impact of initial conditions is analyzed using the difference between hindcast-RIC and hindcast-CIC simulations. While many studies have analyzed the role of initial conditions by two experiments: Ensemble Streamflow Prediction (ESP) and reverse-ESP (revESP). Some discussion regarding the difference between them may be added.

Response: We acknowledge that ESP and revESP experiments have been widely used to evaluate the importance of the initial conditions. ESP simulations sample meteorological forcings from historical years while using "perfect" initial conditions. RevESP samples initial conditions from historical years while using "perfect" meteorological forcings. Comparing ESP and revESP to retrospective simulations yields estimates of the relative importance of initial condition and meteorological forcing. In our manuscripts, we evaluate the role of the initial condition in providing skillful hydrological forecasts within SAHFS-S2S. We compare two sets of hindcast simulations, hindcast with real initial conditions (hindcast-RIC) and hindcast with climatological initial conditions (hindcast-CIC). Although the design of the hindcast-CIC experiment is different from the revESP experiments, the initial conditions of both of the experiments provide no information on year-to-year variations. We think that comparing hindcast-CIC with hindcast-RIC is sufficient to study the role of initial condition in SAHFS-S2S.

We added the following statements in line 399 in the revised manuscript. "The impact of initial condition has also been studied using ESP and revESP methods (Yuan et al., 2016;Shukla and Lettenmaier, 2011;Luo et al., 2008;Ma et al., 2018). These studies have yielded similar conclusions regarding the fact that initial condition has a longer impact when the forecast is initialized in a dry season and a shorter impact when the forecast is initialized in a wet season. "

Minor comments

4. Line 40: The meaning of the sentence "The forecast period . . . in recent years" is not well understood.

Response: We delete this sentence as the sub-monthly to monthly forecast is beyond the scope of the discussion of this paper.

5. Line 44: What is "land component".

Response: We modify the phrase to the "land surface model" in the revised manuscript.

6. Line 46: The sentence "The influence of initial hydrological . . ." is repeated.

Response: We delete one of the sentences.

7. The depth of root zone soil moisture is 1 m?

Response: In Noah-MP, there are four soil layers with thickness varying from 0.1, 0.3, 0.6, and 1 m from the top to the bottom. In this study, we define the depth of root zone soil moisture to be 1 m, which we believe is a reasonable assumption for most of the croplands. We now add the following definition in line 283: "The depth of root zone soil moisture is defined as 1 meter in this study."

8. Some figures, such as Fig. S1, S3, S6, S7, in supplement information are not mentioned in the paper.

Response: We add the following sentences to the manuscript to address this issue. Line 386: The elimination of the initial condition is also observed rather quickly in June in the Mekong basin (Fig. S1) Line 375: Short memories are generally present in areas receiving considerable summer monsoon precipitation, such as the west coast and northeast parts of India, the southern slopes of the Himalayas, and the west coast and the southeast of the Indochinese peninsula (Fig. 8, S3). Line 474: The SAHFS-S2S shows similar skills forecasting drought categories in the Ganges (Fig. S5), Brahmaputra (Fig. S6), and the Indus (Fig. S7) basins.

9. The ESA-CCI SM derived from remote sensing observation has many missing data, so how to deal with the missing data?

Response: We didn't conduct any special treatment for missing data. Rather, to make a fair comparison, when calculating the monthly values from hindcast and retrospective simulations, we only use data at dates when ESA-CCI SM are available as well. We now add this explanation to line 404.

10. In data section, many datasets are used. Here, I suggest that a data table, including data sources, detailed information and variables used, should be listed for easily read.

Response: We now include the Table 1 in the manuscript.

11. Human activities, such as irrigation, has great influence on soil moisture, how to consider irrigation in the NoahMP land surface model?

Response: We thank the reviewer for pointing this out. In the work of Nie et al. (2018,2019), irrigation with source water partitioning has been implemented into Noah-MP along with GRACE data assimilation, which has improved the simulation of water and energy fluxes for High Plains in the US, including surface soil moisture, which is a heavily irrigated region with groundwater depletion issue. We are working to transfer and customize this set up for South Asia and explore how this may impact the forecasting skill.

12. Line 330: Please add a figure of monthly evapotranspiration (ET) for easily explaining.

Response: We now add Fig. S6 to the supplement document and reference Fig. S6 in line 330.

13. Line 780: Figure 7: The difference is computed as correlation between RIC and CIC?

Response: The difference isn't referring to a direct correlation between RIC and CIC. The correlation difference is calculated as the correlation between hindcast-RIC and retrospective simulations minus the correlation between hindcast-CIC and retrospective simulations. This correlation difference denotes the loss of prediction skill without informative initial conditions.

14. In the Case Study of the 2015 South and Southeast Asia section, I suggest a spatial distribution map showing the comparison of drought conditions between retrospective simulation and hindcast ensemble mean may be added.

Response: We'll add the spatial distribution map (Fig. S11, S12) to the supplement. We also add the following texts to line 453. "The forecast of RZSM standardized anomaly generally captures the spatial pattern in the first two months in both the Ganges basin and the Mekong basin (Fig. S11, S12). "

15. Figure 9: the title of right subplots should be (b), (d), (f).

Response: We modify the title of the right subplots to be (b), (d), and (f).

Reference:

Getirana, A., Rodell, M., Kumar, S., Beaudoing, H. K., Arsenault, K., Zaitchik, B., Save, H., and Bettadpur, S.: GRACE Improves Seasonal Groundwater Forecast Initialization over the United States, Journal of Hydrometeorology, 21, 59-71, 2020. Gutmann, E. D., Hamman, J. J., Clark, M. P., Eidhammer, T., Wood, A. W., Arnold, J. R., and Nowak, K.: Evaluating the effect of regional climate inference methodologies in a common framework, In review, 2020. Luo, L., Sheffield, J., and Wood, E.: Towards a global drought monitoring and forecasting capability, 33rd NOAA Annual Climate Diagnostics and Prediction Workshop, 2008, 20-24, Ma, F., Luo, L., Ye, A., and Duan, Q.: Seasonal drought predictability and forecast skill in the semi-arid endorheic Heihe River basin in northwestern China, Hydrology and Earth System Sciences, 22, 5697-5709, 2018. Nie, W., Zaitchik, B. F., Rodell, M., Kumar, S. V., Anderson, M. C., and Hain, C.: Groundwater withdrawals under drought: Reconciling GRACE and land surface models in the United States High Plains Aquifer, Water Resources Research, 54, 5282-5299, 2018. Nie, W., Zaitchik, B. F., Rodell, M., Kumar, S. V., Arsenault, K. R., Li, B., and Getirana, A.: Assimilating GRACE into a Land Surface Model in the presence of an irrigation‐induced groundwater trend, Water Resources Research, 2019. Shukla, S., and Lettenmaier, D.: Seasonal hydrologic prediction in the United States: understanding the role of initial hydrologic conditions and seasonal climate forecast skill, Hydrology and Earth System Sciences, 15, 3529-3538, 2011. Yuan, X., Ma, F., Wang, L., Zheng, Z., Ma, Z., Ye, A., and Peng, S.: An experimental seasonal hydrological forecasting system over the Yellow River basin–Part 1: Understanding the role of initial hydrological conditions, Hydrology and Earth System Sciences, 20, 2437, 2016.

Please also note the supplement to this comment:
https://hess.copernicus.org/preprints/hess-2020-362/hess-2020-362-AC2-

supplement.pdf

[Figure]

**Supplement:**

Table 1: Summary of datasets used in this study. The meteorological forcing fields include precipitation (precip), downward long-wave radiation (LW), downward shortwave radiation (SW), air temperature (Ta), specific humidity (Q), surface air pressure (P), zonal (U), and meridional (V) wind speed.

| Data | Period of Record | Spatial resolution | Temporal resolution | Variable used | Reference |
|---|---|---|---|---|---|
| CHIRPS | 1981-present | 0.05° | 1 day | Precip | Funk et al., 2015 |
| GDAS | 2000-present | 1° × 1° gradually improved to 0.125° × 0.125° | 6 hours | LW, SW, Ta, Q, P, U, V | National Climatic Data Center, 2020 |
| GEOS-S2S-V1 | 1981-2018 Jan | 1°×1.25° | 1 day | Precip, LW, SW, Ta, Q, P, U, V | Borovikov et al., 2017 |
| GEOS-S2S-V2 | 1981-present | 0.5°×0.5° | 1 day | Precip, LW, SW, Ta, Q, P, U, V | Molod et al., 2020 |
| ESA-CCI SM | 1978 Nov - 2019 Dec | 0.25°×0.25° | 1 day | Surface soil moisture | Gruber et al., 2019 |

[Figure]

**Figure S1: Comparison of Air temperature (T), precipitation (Precip), specific humidity (Q), surface pressure (P), solar radiation (SW), longwave radiation (LW), north-south wind speed (NW), east-west wind speed (EW) among Retrospective, raw GEOS-S2S-V1 and downscaled GEOS-S2S-V1 meteorological forcing in the Helmand basin. The root mean squared error normalized by the range of the retrospective meteorological forcing (NRMSE) is shown as the values in each subplot. R denotes the NRMSE between raw GEOS-S2S-V1 and retrospective forcing and D denotes the NRMSE between downscaled GEOS-S2S-V1 and retrospective forcing.**

[Figure]

**Figure S2: The same as Fig. S1 but in the Brahmaputra basin.**

[Figure]

**Figure S3: The same as Fig. S1 but in the Ganges basin**

[Figure]

**Figure S4: The same as Fig. S1 but in the Indus basin**

[Figure]

**Figure S5: The same as Fig. S1 but in the Mekong basin**

[Figure]

**Figure S6. The climatological monthly precipitation and evapotranspiration (ET) in the Indus basin from retrospective and hindcast-RIC simulations.**

[Figure]

**Figure S11. The spatial distribution of monthly RZSM standardized anomaly for 2015 South and Southeast Asia drought calculated from the retrospective simulation**

[Figure]

**Figure S12. The same as Fig. S12 but calculated from ensemble mean of the hindcast-RIC simulation**

---

## Author Response (AR1)

Dear Dr. Yuan,

Thanks for your quick response. We've revised the manuscript as requested and we attach the marked-up version to the end of this response.

Regards,
Yifan Zhou

[revised manuscript text omitted]

10   Supporting information

[Figure]

**Figure S1: Comparison of Air temperature (T), precipitation (Precip), specific humidity (Q), surface pressure (P), solar radiation (SW), longwave radiation (LW), north-south wind speed (NW), east-west wind speed (EW) among Retrospective, raw GEOS-S2S-V1 and downscaled GEOS-S2S-V1 meteorological forcing in the Helmand basin. The root mean squared error normalized by the range of the retrospective meteorological forcing (NRMSE) is shown as the values in each subplot. R denotes the NRMSE between raw GEOS-S2S-V1 and retrospective forcing and D denotes the NRMSE between downscaled GEOS-S2S-V1 and retrospective forcing.**

[Figure]

20 **Figure S2: The same as Fig. S1 but in the Brahmaputra basin.**

[Figure]

**Figure S3: The same as Fig. S1 but in the Ganges basin**

[Figure]

Figure S4: The same as Fig. S1 but in the Indus basin

[Figure]

**Figure S5: The same as Fig. S1 but in the Mekong basin**

[Figure]

35  **Figure S6: Interannual correlation of precipitation between retrospective simulations (i.e. CHIRPS) and hindcasting simulations (i.e. downscaled GEOS-S2S-V1 precipitation) which is computed using precipitation data from year 2000 to 2017 in months May (1-month lead time) to January (9-month lead time) (subplot (a) to subplot (i)). The hatches denote the areas with statistically significant correlation at 0.95 confidence level.**

[Figure]

**Figure S7. The climatological monthly precipitation and evapotranspiration (ET) in the Indus basin from retrospective and hindcast-RIC simulations.**

[Figure]

**Figure S8:** **The largest lead month when the difference of RZSM forecast skills between hindcast-RIC and hindcast-CIC against retrospective simulations still remains statistically significant.**

[Figure]

**Figure S9: Comparison between retrospective simulations and hindcast simulations in the Mekong basin of (a) monthly time series of precipitation, (b) monthly climatology of root zone soil moisture (RZSM), (c) inter-annual correlations of basin-averaged RZSM standardized anomaly and precipitation, and (d) monthly standardized anomaly of RZSM (the red error bars represent for the standard deviation of separate hindcast ensemble members). Please note that the time axis for monthly time series are rearranged so that the data for the same month is grouped.**

[Figure]

**Figure S10**: Same as Fig. S9, but for the Brahmaputra basin.

60

[Figure]

Figure S11. The spatial distribution of monthly RZSM standardized anomaly for 2015 South and Southeast Asia drought calculated from the retrospective simulation

[Figure]

**Figure S12. The same as Fig. S11 but calculated from ensemble mean of the hindcast-RIC simulation**

70

[Figure]

**Figure S13: Fractions of areas of different drought categories in Ganges basin calculated using RZSM data from retrospective and hindcast-RIC simulations from (a) May to (i) January.**

75

[Figure]

**Figure S14: Same as Fig. S13 but for the Brahmaputra basin.**

80

[Figure]

**Figure S15: Same as Fig. S13 but for the Indus basin.**

85